**EMBO** *reports*

# Viral oncogene EBNALP regulates YY1 DNA binding and alters host 3D genome organization

Chong Wang [1,2,5 ✉], Merrin Manlong Leong [1,5], Weiyue Ding [1,5], Yohei Narita[1,5], Xiang Liu[3], Hongbo Wang[1], Stefanie P T Yiu[1], Jessica Lee[1], Katelyn R S Zhao[1], Amy Cui[1], Benjamin Gewurz[1], Wolfgang Hammerschmidt [4], Mingxiang Teng [3 ✉] & Bo Zhao [1 ✉]

## Abstract

The Epstein–Barr virus (EBV) nuclear antigen leader protein (EBNALP) is essential for the immortalization of naive B lymphocytes (NBLs). However, the mechanisms remain elusive. To understand EBNALP's role in B-cell transformation, we compare NBLs infected with wild-type EBV and an EBNALP-null mutant EBV using multi-omics techniques. EBNALP inactivation alters enhancer–promoter interactions, resulting in decreased CCND2 and increased CASP1 and BCL2L11 expression. Mechanistically, EBNALP interacts with and colocalizes with the looping factor YY1. Depletion of EBNALP reduces YY1 DNA-binding and enhancer–promoter interactions, similar to effects observed with YY1 depletion. Furthermore, EBNALP colocalizes with DPF2, a protein that binds to H3K14ac and H4K16ac. CRISPR depletion of DPF2 reduces both EBNALP and YY1 DNA binding, suggesting that the DPF2/EBNALP complex may tether YY1 to DNA to increase enhancer–promoter interactions. EBNALP inactivation also increases enhancer–promoter interactions at the CASP1 and BCL2L11 loci, along with elevated DPF2 and YY1 binding and DNA accessibility. Our data suggest that EBNALP regulates YY1 to rewire the host genome, which might facilitate naive B-cell transformation.

**Keywords** Epstein–Barr Virus; EBNALP; YY1; HiChIP; 3D Genome Organization
**Subject Category** Chromatin, Transcription & Genomics

## Introduction

Within the nucleus, the genomic DNA is organized in an extremely complicated yet ordered structure, allowing efficient interactions between remote enhancers/silencers and their direct target gene promoters (Bonev and Cavalli, 2016). Looping factors CTCF, YY1, and cohesin subunits RAD21, SMC1, and SMC3, play important roles in organizing the genome, enabling efficient interactions between remote enhancers and promoters and looping out intervening sequences (van Ruiten and Rowland, 2018). CTCF binds insulators between topologically associated domains (TADs), allowing efficient enhancer–promoter interactions within TADs (Nanni et al, 2020). CTCF can form dimers to bring remote DNA they are binding to close proximity. YY1 is a DNA-binding protein that can also dimerize and preferentially binds to enhancers and promoters within TADs (Li et al, 2021; Wang et al, 2022; Weintraub et al, 2017). YY1 dimerization positions enhancers and promoters in close proximity, facilitating remote enhancers and promoters to form active transcription complexes with other basal and activated transcription factors (TFs). CRISPR disruption of YY1 binding sites interferes with the interaction between enhancer and promoter. Inducible YY1 degradation results in global enhancer–promoter interaction changes, and the defects can be rescued by tethering dCAS9-YY1 to DNA (Weintraub et al, 2017). Cohesin subunits form a ring and wrap around the CTCF or YY1 dimers to stabilize the DNA interactions. Extensive interactions between looping factors have been reported (Donohoe et al, 2007; Stedman et al, 2008). Looping factors frequently interact with each other, and the interactions can further strengthen the enhancer–promoter interactions.

3D genome reorganization occurs during development (Dixon et al, 2012; Phillips-Cremins et al, 2013). Mutations in looping factor family members such as cohesin loading factor NIPBL1 can lead to developmental defects (Cornelia de Lange syndrome) (Panarotto et al, 2022). In total, 36% of the TADs altered during differentiation from embryonic stem cells to four different lineages (Dixon et al, 2012). During B-cell differentiation, the naive B-cell genome undergoes extensive rewiring in the germinal center that allows transcription reprogramming (Bunting et al, 2016). TAD merger and enhancer network formation allowed active epigenetic regulation of BCL6 which is critically important for GC cell function (Bunting et al, 2016). Oncogenic mutations, gene rearrangements, deletion, or amplification all can lead to altered genome organization and lead to deregulated oncogene expression (Corces and Corces, 2016). Genome architectural changes are profound in cancers (Johnstone et al, 2020). Methylation of CTCF sites at TAD boundaries prevents CTCF DNA binding and leads to aberrant oncogene expression (Flavahan et al, 2016). CTCF and cohesin binding sites are also frequently mutated (Katainen et al, 2015).

[1]Division of Infectious Disease, Department of Medicine, Brigham and Women's Hospital, Harvard Medical School, Boston, MA, USA. [2]Department of Diagnostic and Biological Sciences, School of Dentistry, University of Minnesota, Minneapolis, MN 55455, USA. [3]Department of Biostatistics and Bioinformatics, H. Lee Moffitt Cancer Center and Research Institute, Tampa, FL 33612, USA. [4]Research Unit Gene Vectors, Helmholtz Zentrum München, German Research Center for Environmental Health and German Center for Infection Research, Munich, Germany. [5]These authors contributed equally: Chong Wang, Merrin Manlong Leong, Weiyue Ding, Yohei Narita. ✉E-mail: wan01642@umn.edu; mingxiang.teng@moffitt.org; bzhao@bwh.harvard.edu

 

Viral infection can also cause host genome reorganization. Influenza A virus NS1 protein can cause global transcription readthrough that disrupts cohesin and CTCF-mediated looping (Heinz et al, 2018). EBV infection of B cells also globally alters host genome organization (Wang et al, 2023). Viral infection can induce enhancer–promoter looping. Epstein–Barr virus (EBV) nuclear antigen 2 (EBNA2) can form enhancers 400–500 kb upstream of *MYC* and loop to the *MYC* promoter to activate gene expression (Jiang et al, 2017). Looping factors are also important for viral gene expression (McClellan et al, 2013). CTCF can regulate the gene expression of several different viruses, such as EBV, KSHV, and HPV (Chen et al, 2014; Chen et al, 2012; Tempera et al, 2010). CTCF and cohesin are also important in EBV latency-type switch (Morgan et al, 2022; Tempera et al, 2011).

EBV is causally associated with ~200,000 cases of various cancers each year. These cancers include Burkitt's lymphoma, Hodgkin's disease, AIDS lymphomas, post-transplant lymphoproliferative disease (PTLD), nasopharyngeal carcinoma, and 10% of gastric cancers (Cohen et al, 2011). In vitro, EBV infection of human primary resting B lymphocytes (RBLs) transforms them into lymphoblastoid cell lines (LCLs). In LCLs, a viral program termed type III latency ensures the expression of six EBV nuclear antigens (EBNAs), three latent membrane proteins (LMPs), small RNAs EBER, and many micro-RNAs. In immunocompromised patients suffering from PTLD or AIDS, lymphoma cells express the identical set of viral genes. LCLs are therefore ideal model system to study EBV oncogenic transformation. Among the viral latency genes, EBNA1 tethers EBV episomes to the host chromosome during mitosis to enable EBV episome persistence (De Leo et al, 2020). EBNA2 is a transcription factor that actives the expression of viral and host gene expression, including MYC (Kaiser et al, 1999). EBNA3A/3C repress *CDKN2A* expression to overcome senescence (Maruo et al, 2005; Maruo et al, 2011). LMP1 activates NF-κB (Laherty et al, 1992).

EBNA leader protein (EBNALP) is required for EBV to immortalize naive B lymphocytes (NBLs) (Mannick et al, 1991; Pich et al, 2019; Szymula et al, 2018). Functionally, EBNALP strongly coactivates EBNA2 in reporter assays (Harada and Kieff, 1997; Nitsche et al, 1997) via its interaction with the transactivation domain of EBNA2 (Peng et al, 2004). EBNALP transcription coactivation is supported by EP300, modulation of SP100 localization (Ling et al, 2005), and removal of transcription repressors from EBNA2 (Portal et al, 2011; Wang et al, 2018). EBNALP also modulates RNA splicing (Manet et al, 2021). Importantly, genome-wide, 33% of EBNALP ChIP-seq peaks map to cellular promoters compared to only 14% of all EBNA2 peaks (Portal et al, 2013; Zhao et al, 2011). EBNALP binding sites are highly marked by H3K4me3, H3K27ac and other active histone marks in both LCLs and RBLs (Portal et al, 2013). Similarly, YY1 also preferentially binds to promoter regions (Portal et al, 2013; Weintraub et al, 2017).

We report here that EBNALP manipulates YY1 DNA binding to alter host 3D genome organization, enabling cell cycle progression and preventing cell death.

# Results

## EBNALP regulates host gene expression

EBNALP is a highly spliced viral gene composed of multiple identical repeats of W exons followed by unique Y1 and Y2 exons.

EBV mutants with stop codons inserted into each of the W repeats have been established (Pich et al, 2019; Szymula et al, 2018). We used RNA-seq to identify EBNALP-regulated genes. The wild-type (wt) or mutant EBV virus were released into the supernatant of HEK293 cells harboring wt EBV genomes or EBNALP knockout (LPKO) EBV, respectively, after induction of EBV's lytic replication. Virus supernatants were concentrated by centrifugation and titered using Daudi cells to adjust virus doses for infecting NBL from healthy donors. Cells were harvested 2 days after infection. >80% of the cells were GFP positive by FACS analyses (Fig. EV1A,B). Western blotting was used to determine the expression of EBNA2 and EBNALP. Wt and LPKO virus-infected cells expressed similar amount of EBNA2 while EBNALP was detected only in wt virus-infected cells but not in knockout virus-infected cells (Fig. EV1C). Most of the cells were in the G1 phase (Fig. EV1D) as expected (Pich et al, 2019) at this stage. Wt virus-infected cells grew into LCLs while LPKO virus-infected NBLs failed to grow into LCLs (Fig. EV1E). Total RNAs were extracted from NBLs 2 days after wt or LPKO virus infection in triplicates and analyzed by RNA-seq. DESeq2 was used to normalize gene read counts and to identify differentially expressed genes in wt or LPKO virus-infected cells (Love et al, 2014). In total, 485 genes were upregulated at least twofold in wt EBV-infected cells, including *CCND2*, *BIRC3*, and *MIR155HG*, with adjusted $q$ value < 0.05. Overall, 375 genes were upregulated in LPKO virus-infected cells including *CASP1*, *BCL2L11(BIM)*, and *IFNGR1*, with adjusted $q$ value < 0.05 (Figs. 1A and EV1F–I). Pathway analysis of all genes differentially regulated by EBNALP, both up and downregulated, found intestinal immune network for IgA production, viral protein interaction with cytokine and cytokine receptor, NF-kB signaling pathway, and some additional pathways were enriched (Fig. 1B,C).

## EBNALP globally reorganizes host 3D genome architecture

In all, 59% of EBNALP ChIP-seq peaks overlap with YY1 in LCLs, suggesting that EBNALP may contribute to LCL 3D genome organization (Portal et al, 2013). To determine if EBNALP alters host 3D genome organization, H3K27ac HiChIP, an assay that combines genome-wide chromosome interaction capture and H3K27ac ChIP selecting enhancer interactions was used (Lieberman-Aiden et al, 2009; Mumbach et al, 2016). HiChIP-identified loops mostly overlap with loops identified by chromosome conformation capture followed by deep sequencing (HiC) (Lieberman-Aiden et al, 2009; Mumbach et al, 2017). NBLs were infected with wt and LPKO EBV for 48 h. Cells were first crosslinked, and DNA was cut by MboI. DNA ends were filled with biotinylated dATP (in the presence of remaining nucleotide triphosphates) and ligated. ChIP was used to select the ligation products enriched with H3K27ac, and the ligation products were captured using avidin-coupled beads. The captured DNA was paired-end sequenced and the sequencing reads were mapped to the human genome (hg19). Significant intra-chromosomal interactions were called using hichipper. Diffloop was used to identify loops altered following EBNALP inactivation (Appendix Fig. S1A). In the absence of EBNALP, 4141 loci showed reduced intra-chromosomal interactions (operationally termed "EBNALP induced contacts"), and 5079 loci showed increased intra-chromosomal interactions (termed "EBNALP repressed contacts") (Appendix Fig. S1B). In all, 39,699 interactions

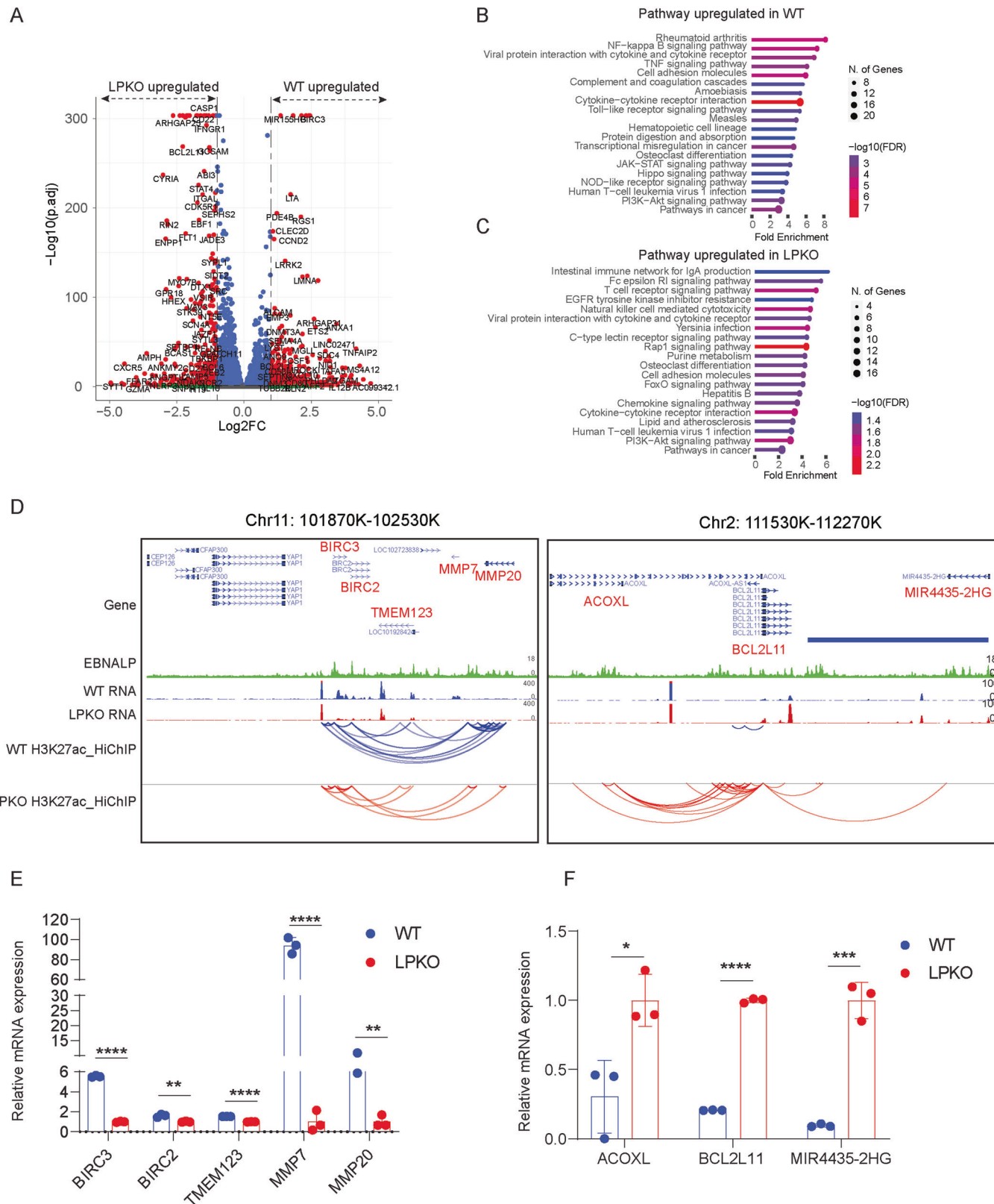

**Figure 1. EBNALP alters genome organization to regulate gene expression.**

(A) EBNALP-regulated genes. RNA-seq analysis of NBLs infected with wt or LPKO EBV for 2 days is shown in volcano plot ($-\log 10$ $q$ value vs log2 mRNA fold change) ($n = 3$ biological replicates). Differential gene were obtained using DESeq2 analysis. (B, C) KEGG pathway analysis of differentially expressed genes (fold change >2, $q$ value < 0.05). (D) EBNALP alters genome organization. NBLs infected with wt or LPKO EBV for 2 days were analyzed by H3K27ac HiChIP. EBNALP ChIP-seq, RNA-seq and H3K27ac HiChIP tracks are shown at the BIRC3 and BCL2L11 loci. Gene expression significantly affected by EBNALP inactivation are highlighted in red. HiChIP was performed in $n = 2$ biological replicates. (E) mRNA levels of genes highlighted in (C) from the RNA-seq data. Statistical significance was tested between the WT and LPKO groups ($P$ value less than 0.0001 is shown as <0.0001. For BIRC3, $P < 0.0001$; BIRC2, $P = 0.003$; TMEM123, $P < 0.0001$; MMP7, $P < 0.0001$; MMP20, $P = 0.0075$). (**$P < 0.01$, ****$P < 0.0001$). (F) mRNA levels of genes highlighted in (C) from the RNA-seq data. Statistical significance was tested between the WT and LPKO groups ($P$ value less than 0.0001 is shown as <0.0001. For ACOXL, $P = 0.020$; BCL2L11, $P < 0.0001$; MIR4435-2HG, $P = 0.0003$). (*$P < 0.05$, ***$P < 0.001$, ****$P < 0.0001$). A two-tailed unpaired $t$ test was used for RNA-seq individual gene statistical analyses. The error bars indicate the SD for the averages of $n = 3$ biological replicates. Source data are available online for this figure.

were not affected by EBNALP (stable contacts) (Appendix Fig. S1B). The distance between two genomic interactions varied greatly, with the longest interactions between genomic regions ~2000 kb apart (Appendix Fig. S1C). For stable contacts, the most frequent interactions were between 500 and 1000 kb (Appendix Fig. S1D). For induced or repressed contacts, the most frequent interactions were between 100 and 500 kb. The proportions of enhancer–enhancer, enhancer–promoter, and promoter–promoter interactions were similar among stable, induced or repressed contacts. ~50% of contacts had nearby EBNALP peaks at both contact points in all three groups with different contact patterns. ~25% of contacts had EBNALP peaks at one of the contact points. For example, EBNALP inactivation significantly decreased intragenic interactions at the *BIRC3* locus and increased intragenic interactions at the *BCL2L11* locus (Fig. 1D). BIRC3 is essential for LCL growth and survival. CRISPR depletion significantly reduces LCL growth. BCL2L11 is known to be repressed by EBNA3C (Paschos et al, 2012; Wood et al, 2016). EBNALP represses BCL2L11 early during infection, as EBNA3C is expressed later following EBV infection (Nikitin et al, 2010). It is known that *BIRC3* transcription is significantly repressed already 48 h after infection of naive B cells with wild-type EBV whereas transcription of *BCL2L11* is induced after 24 h to become repressed later starting at day 3 post infection. Both genes are differentially expressed when comparing wt versus LPKO EBV infected B cells 48 h post infection (Fig. 1E,F) reflecting the loss (*BIRC3*) and gain (*BCL2L11*) of intragenic interactions. These data indicate that EBNALP can extensively alter the host enhancer–promoter interaction to regulate the expression of genes linked to B-cell reprogramming and survival.

## DPF2 recruits EBNALP to DNA

It is not known how EBNALP is tethered to chromatin. To address this question, we generated LCLs expressing HA-FLAG tagged EBNALP to investigate its role in a stably EBV-transformed and established B-cell line. HA antibody was used to immune precipitate EBNALP together with associated proteins. LCLs expressing non-tagged EBNALP was used as a control. The purified proteins were analyzed by liquid chromatography-mass spectrometry (LC-Mass spec). DPF2 was identified as an EBNALP interacting protein while EBNA2 was not immune precipitated. DPF2 is a component of the SWI-SNF chromatin remodeling complex (Bogershausen and Wollnik, 2018). DPF2 associates with chromatin through binding to H3K14ac and H4K16ac (Huber et al, 2017). We compared EBNALP ChIP-seq peaks with ENCODE GM12878 LCL, an ENCODE tier 1 cell line, DPF2 ChIP-seq peaks. In all, 82% of all EBNALP peaks overlapped with DPF2 peaks (Fig. 2A). When EBNALP peaks overlapped with DPF2

peaks, the EBNALP ChIP-seq signals were much higher than peaks lacking DPF2 binding (Fig. 2B). In LCLs, EBNALP colocalized with DPF2 by immune fluorescence (IF), in agreement with the ChIP-seq data (Fig. 2C). Immune precipitation was used to further evaluate EBNALP DPF2 interactions. Anti EBNALP antibody JF186 immune precipitated EBNALP efficiently from IB4 LCLs, a much older LCL with stable EBNALP size (Fig. 2D). EBNALP co-precipitated HA95, a known EBNALP interacting protein. EBNALP also co-precipitated DPF2 (Fig. 2D), supporting EBNALP DPF2 association. To further determine if EBNALP is tethered to chromatin by DPF2, CRISPR depletion was used. DPF2 and EBNALP mostly overlapped at *CCND2*, *CD21*, and *HES1* enhancers or promoters. (Fig. 2E). IB4 LCLs stably expressing CAS9 were transduced with lentivirus expressing sgRNA targeting DPF2. After puromycin selection, Western blotting was used to determine the depletion efficiency. CRISPR depletion reduced DPF2 expression by ~80% (Fig. 2E). EBNALP ChIP followed by qPCR was used to determine the EBNALP DNA binding in non-targeting sgRNA treated control cells or DPF2-depletion cells. Whereas control antibody precipitated minimal amount of DNA in both control and DPF2-depletion cells, the JF186 antibody precipitated ~0.1%–0.15% of input DNA in control cells. DPF2 depletion significantly reduced EBNALP DNA binding ($P < 0.01$) (Fig. 2E), suggesting that DPF2 may tether EBNALP to chromatin.

## EBNALP regulates DPF2 DNA binding

DPF2 CUT&RUN was used to examine DPF2 DNA binding in NBLs infected with wt or LPKO virus for 48 h (Appendix Fig. S2A,B). EBNALP inactivation decreased DPF2 DNA binding at the *PEE4B* and AL365434.1 loci, accompanied by reduced enhancer–promoter looping (Fig. 3A) and gene expression (Fig. 3B). At wt EBV unique DPF2 loci, more HiChIP loops in wt EBV-infected cells were enriched than LPKO EBV-infected cells (Fig. 3C, left). At LPKO EBV unique DPF2 loci, more HiChIP loops were enriched in LPKO virus-infected cells than wt EBV-infected cells (Fig. 3C, middle). No difference was seen at random control regions without DPF2 binding (Fig. 3C, right).

## EBNALP regulates YY1 binding to enhancers and promoters

EBNALP ChIP-seq peaks greatly overlap with YY1 and CTCF peaks (Portal et al, 2013) which led us to ask if EBNALP might affect YY1 DNA binding. In NBLs infected with wt or LPKO EBV for 48 h, we performed YY1 CUT&RUNs. EBNALP inactivation resulted in a loss of 1047 and a gain of 1084 significant YY1 peaks whereas 20,733 peaks were unchanged (Fig. EV2A,B). *CCND2*

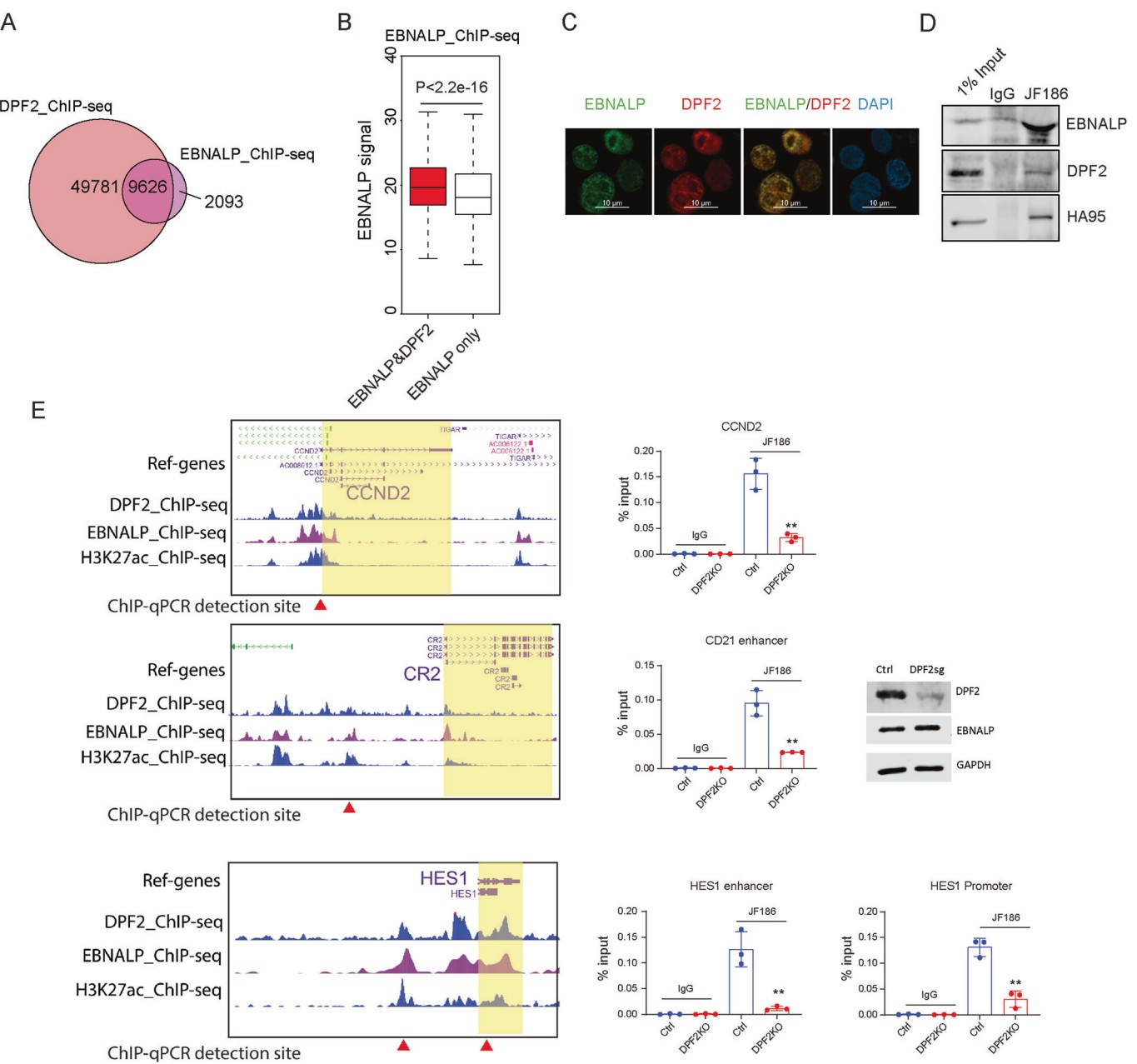

**Figure 2. DPF2 recruits EBNALP to DNA.**

(A) LCL DPF2 ChIP-seq peaks overlap with LCL EBNALP ChIP-seq peaks. (B) Normalized EBNALP ChIP-seq signals (mean coverage) at sites with overlapping DPF2 binding and sites lacking DPF2 binding. Boxplot plot: center value is the medium; upper and lower bounds of boxes are upper and lower quartile, respectively; whiskers extend by 1.5*(upper quartile–lower quartile). $P$ value was calculated using Wilcoxon rank-sum test. EBNALP ChIP-seq was from $n = 2$ biological replicates. (C) Immunofluorescence staining EBNALP (green), DPF2 (red) and DAPI (blue) in NBLs infected with wt B95.8 EBV for 7 days. Immunofluorescence is representative of $n = 3$ biological replicates. (D) EBNALP associates with DPF2 in IB4 LCLs. EBNALP antibody JF186 was used to immune precipitate EBNALP. Co-precipitated proteins were examined by Western blotting. Blots are representative of $n = 2$ biological replicates. (E) Depletion of DPF2 reduces EBNALP DNA binding. DPF2, EBNALP, and H3K27ac ChIP-seq tracks at CCND2, CR2, and HES1 loci are shown (gene bodies are indicated in yellow shade). DPF2 was depleted using CRISPR from IB4 LCLs (western blot). EBNALP ChIP-qPCR was used to determine the effect of DPF2 depletion on EBNALP DNA binding. The red arrows under the ChIP-seq tracks indicate sites for qPCR amplification. IgG was used as a negative control. Depleting DPF2 significantly decreases EBNALP DNA binding at all loci evaluated. Statistical significance was tested between Ctrl and DPF2 knockout groups pulled down by JF186. (For CCND2, $P = 0.0022$; CD21 enhancer, $P = 0.0027$; HES1 promoter, $P = 0.0018$; HES1 enhancer, $P = 0.0046$) (**$P < 0.01$). A two-tailed unpaired $t$ test was used for ChIP-qPCR statistical analyses. The error bars indicate the SD for the averages of $n = 3$ biological replicates. Source data are available online for this figure.

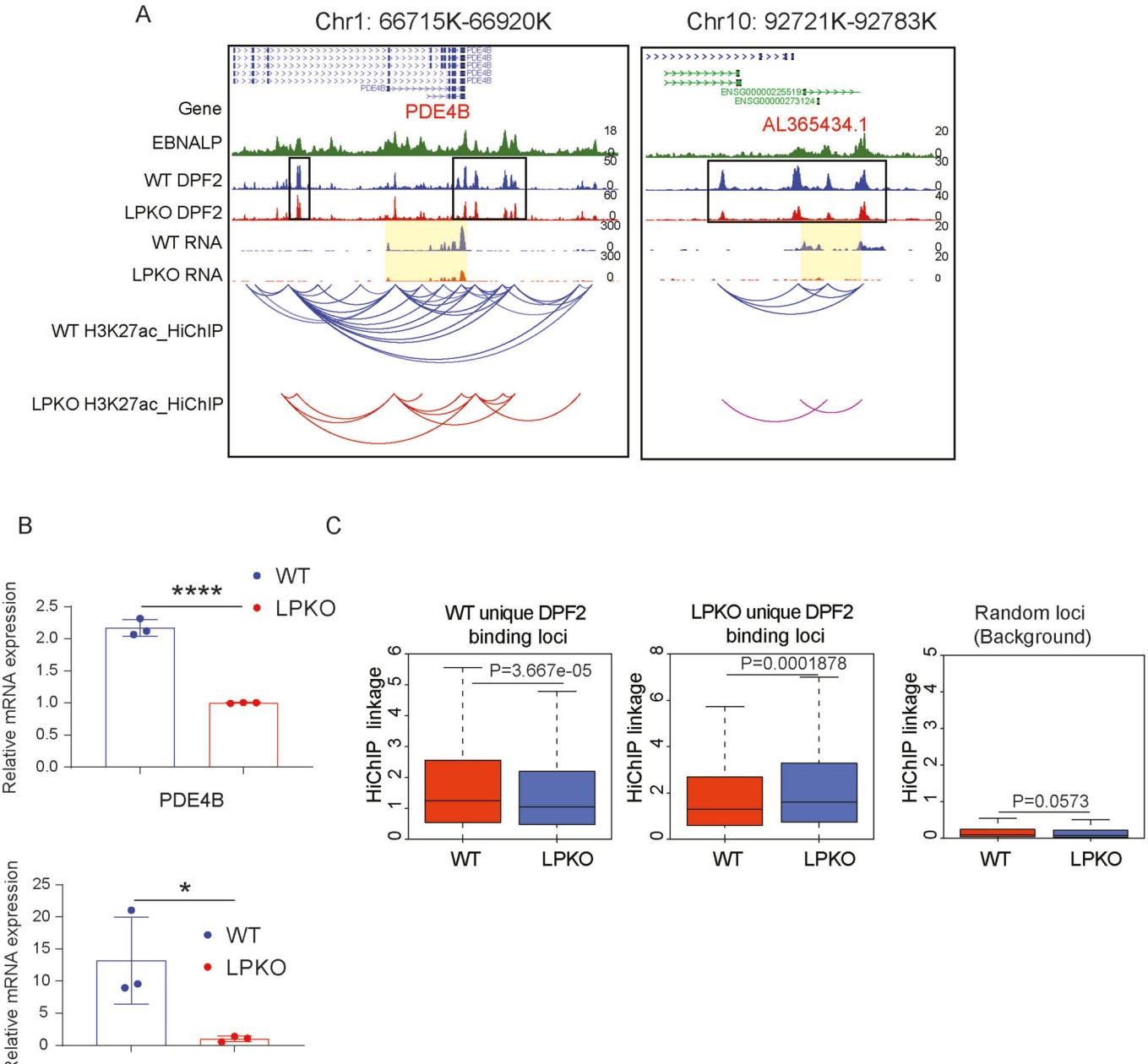

Figure 3. EBNALP increases DPF2 DNA binding.

(A) EBNALP increased enhancer–promoter interactions correlated with increased DPF2 DNA binding. Tracks for EBNALP ChIP-seq, DPF2 CUT&RUN, RNA-seq and H3K27ac HiChIP from wt and LPKO EBV-infected NBLs around PDE4B and long non-coding RNA AL365434.1 loci are shown. DPF2 differential peaks are indicated by black boxes. PDE4B and AL365434.1 gene bodies are highlighted in yellow. DPF2 CUT&RUN was performed in $n = 2$ biological replicates. (B) PDE4B and AL365434.1 mRNA levels from (C). Statistical significance was tested between the WT and LPKO groups (P value less than 0.0001 is shown as <0.0001. For PDEB4, $P < 0.0001$; AL365434.1, $P = 0.0364$). (*$P < 0.05$, ****$P < 0.0001$). A two-tailed unpaired $t$ test was used for RNA-seq individual gene statistical analyses. The error bars indicate the SD for the averages of $n = 3$ biological replicates. (C) H3K27ac HiChIP interactions associated to the differentially enriched DPF2 binding sites between WT (left), LPKO (middle), and control region (right). H3K27Ac HiChIP signals were estimated using HiChIP read pileups from both WT (red) and LPKO (blue) conditions. Sites are defined to include the differential DPF2 peaks as well as their nearby flanking regions (3 kb from both sides) to account for potential mismatch of the peak centers between DPF2 binding and H3K27Ac. Control regions were randomly generated with the number and width match the DPF2 binding sites. Boxplot plot: center value is the medium; upper and lower bounds of boxes are upper and lower quartile, respectively; whiskers extend by 1.5*(upper quartile–lower quartile). P value was calculated using Wilcoxon rank-sum test. HiChIP and DPF2 CUT&RUN were from $n = 2$ biological replicates. Source data are available online for this figure.

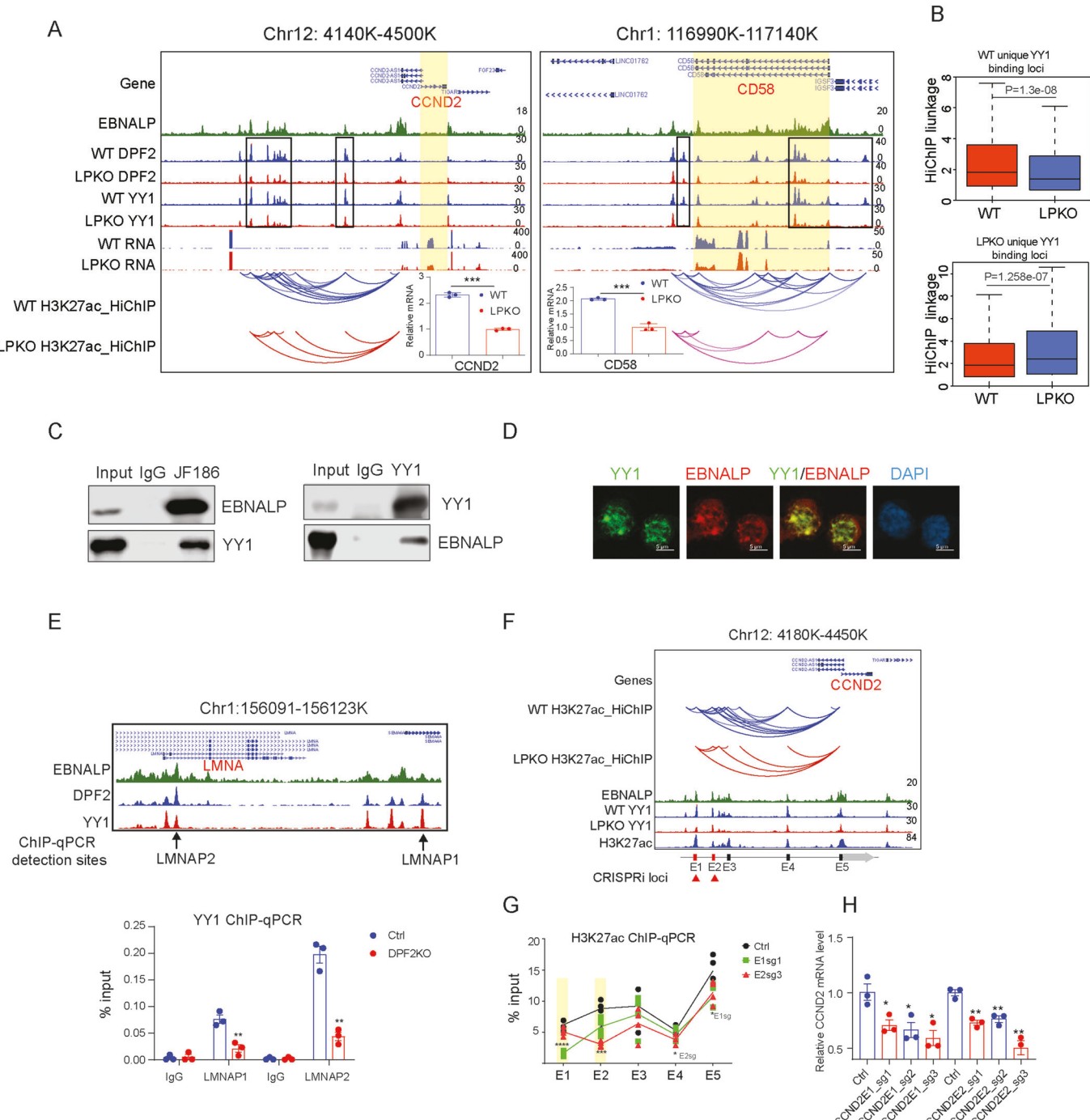

encoding cyclin D2 is the first EBNALP-induced gene identified, is maximally expressed on day 3 post infection and is essential for LCL establishment and growth (Fig. EV2C) (Ma et al, 2017; Mrozek-Gorska et al, 2019; Sinclair et al, 1994). An enhancer cluster ~200 kb upstream of *CCND2* looped to its transcriptional start site (TSS) (Fig. 4A, left). In cells infected with the LPKO EBV, looping between this enhancer and TSS was reduced accompanied with reduced transcription of the gene by a factor of ~2.1. EBNALP inactivation also reduced DPF2 binding to the enhancers. EBNALP

inactivation significantly reduced YY1 binding to the enhancers (Fig. 4A; Appendix Fig. S4D), but YY1 binding at the *CCND2* promoter was not affected by EBNALP inactivation. Reduced enhancer looping to TSS correlated with reduced YY1 binding, suggesting that EBNALP may increase *CCND2* expression via this enhancer–promoter interaction. At the *CD58* locus, EBNALP inactivation reduced enhancers looping to its promoter, consistent with reduced DPF2 and YY1 DNA binding at enhancers and promoter (Fig. 4A, right). Together, these data suggested that

◀ **Figure 4. EBNALP recruits YY1 to DNA to regulate genome organization and gene expression.**

(A) EBNALP induces YY1 DNA binding to sites with increased enhancer–promoter interactions. EBNALP ChIP-seq, DPF2, YY1 CUT&RUNs, RNA-seq and H3K27ac HiChIP from wt or LPKO virus-infected NBLs at CCND2 and CD58 loci are shown. Regions with increased YY1 binding are indicated by boxes. Gene bodies are indicated by yellow. RNA-seq data are shown within the boxes. Statistical significance was tested between the WT and LPKO groups (for CCND2, $P = 0.0001$; CD58, $P = 0.0003$) (***$P < 0.001$). A two-tailed unpaired $t$ test was used for RNA-seq individual gene statistical analyses. The error bars indicate the SD for the averages of $n = 3$ biological replicates. YY1 CUT&RUN was performed in $n = 2$ biological replicates. (B) HiChIP linkages at wt specific YY1 binding site in wt EBV-infected cells and LPKO specific YY1 binding sites in LPKO EBV-infected cells. Boxplot plot: center value is the medium; upper and lower bounds of boxes are upper and lower quartile, respectively; whiskers extend by 1.5* (upper quartile–lower quartile). $P$ values were calculated using Wilcoxon rank-sum test. HiChIP and YY1 CUT&RUN were from $n = 2$ biological replicates. (C) Co-immunoprecipitation of EBNALP and YY1 from IB4 LCLs. JF186 and YY1 antibodies were used to precipitate EBNALP (left) and YY1 (right), respectively. YY1 and anti-EBNALP 4D3 antibodies were used in western blot. 1% input was included in blotting. Blots are representative of $n = 2$ biological replicates. (D) Immunofluorescence staining of YY1 (green), EBNALP (red), and DAPI (blue) after B95.8 EBV infection of NBLs for 7 days. Immunofluorescence are representative of $n = 2$ biological replicates. (E) Depletion of DPF2 reduces YY1 DNA binding. LCL EBNALP ChIP-seq, DPF2 and YY1 CUT&RUN from wt EBV-infected NBLs around LMNA locus are shown on the top. The black arrows indicate the region selected for qPCR detection in ChIP assay. DPF2 was depleted by CRISPR. YY1 DNA binding was determined by ChIP-qCPR. Statistical significance was tested between Ctrl and DPF2 knockout groups pulled down by YY1 antibody. (For LMNAP1, $P = 0.0064$; LMNAP2, $P = 0.0010$). A two-tailed unpaired $t$ test was used for ChIP-qPCR statistical analyses. The error bars indicate the SEM for the averages of $n = 3$ biological replicates (**$P < 0.01$). (F) CRISPRi perturbation of CCND2 enhancers. The positions of sgRNAs and qPCR primers are shown under the tracks. (G) CRISPRi reduces H3K27ac signals at CCND2 enhancer and promoter. H3K27ac ChIP-qPCR was done following CRISPRi perturbation using primers indicated in (G). Statistical significance was tested between Ctrl and E1sg or E2sg2 groups. ($P$ value < 0.0001 was shown as <0.0001. For E1 locus, Ctrl vs E1sg, $P < 0.0001$; E2 locus, Ctrl vs E1sg, $P = 0.0193$; Ctrl vs E2sg, $P = 0.0001$; E4 locus, Ctrl vs E2sg, $P = 0.0191$; E5 locus, Ctrl vs E1sg, $P = 0.0360$). A two-tailed unpaired $t$ test was used for ChIP-qPCR statistical analyses. The error bars indicate the SEM for the averages of $n = 4$ biological replicates. (*$P < 0.05$, ***$P < 0.001$, ****$P < 0.0001$). (H) CRISPRi at CCND2 enhancers reduce CCND2 expression. For each locus, three sgRNAs were designed. qRT-PCR was used to measure CCND2 expression following CRISPRi. CCND2 transcription from the control sgRNA group was set to 1. Statistical significance was tested between Ctrl and different sgRNA groups. (for CCND2E1_sg1, $P = 0.0286$; CCND2E1_sg2, $P = 0.0279$; CCND2E1_sg3, $P = 0.0154$; CCND2E2_sg1, $P = 0.0018$; CCND2E2_sg2, $P = 0.0042$; CCND2E2_sg3, $P = 0.0021$). A two-tailed unpaired $t$ test was used for ChIP-qPCR statistical analyses. The error bars indicate the SEM for the averages of $n = 3$ biological replicates (*$P < 0.05$, **$P < 0.01$). Source data are available online for this figure.

EBNALP may alter the looping factor YY1 DNA binding to affect enhancer–promoter interactions. Genome-wide, in wt EBV-infected cells, unique YY1 binding sites had significantly more HiChIP links between enhancers–promoters than unique YY1 sites in LPKO EBV infected cells (Fig. 4B, top). In LPKO virus-infected cells, unique YY1 binding sites had significantly more HiChIP links than unique YY1 sites in wt EBV-infected cells (Fig. 4B, bottom).

## EBNALP associates with YY1 in LCLs

EBNALP and YY1 ChIP-seq peaks significantly overlap. Since some of the YY1 binding sites were only observed in wt EBV but not in mutant EBNALP EBV-infected B cells, we hypothesized that EBNALP might recruit YY1 to EBNALP sites. To determine if EBNALP interacts with YY1 in LCLs, reciprocal immune precipitations were used. The EBNALP-specific JF186 antibody efficiently precipitated endogenous YY1 together with EBNALP in LCLs (Fig. 4C). Reciprocally, a YY1-specific antibody co-precipitated EBNALP in LCLs detected by anti-EBNALP antibody 4D3 (Fig. 4C). These data indicated that EBNALP and YY1 associate in LCLs at physiological expression levels. To further confirm this result, IF was used to evaluate the EBNALP and YY1 localization in NBLs infected with wt EBV for 7 days. EBNALP and YY1 signals significantly overlapped in newly EBV-infected NBL nuclei (Fig. 4D). These data suggest that EBNALP colocalizes and associates with YY1 in LCLs.

## DPF2 CRISPR depletion reduces YY1 binding to DNA

Since DPF2 tether EBNALP to chromatin, it is possible that in the absence of *DPF2*, cells also show a reduced YY1 DNA binding. To test this hypothesis, we depleted *DPF2* and analyzed the multiple enhancer sites at the *LMNA* locus with strong EBNALP, DPF2, and YY1 peaks (Fig. 4E). YY1 ChIP-qPCRs were used to determine YY1 binding to the indicated enhancers (upwards arrows, Fig. 4E) following DPF2 depletion. DPF2 depletion reduced YY1 DNA binding ($P < 0.01$) compared with LCLs treated with non-targeting

sgRNAs (Fig. 4E), suggesting that DPF2 is also important for YY1 DNA binding at these loci.

## CRISPRi perturbation of *CCND2* enhancers reduces H3K27ac signals and *CCND2* expression

CRISPRi was used to further determine the functional significance of *CCND2* enhancers and transcription of this gene. LCLs stably expressing dCAS9-KRAB-MeCP2 fusion protein were transduced with lentiviruses expressing sgRNAs targeting two *CCND2* enhancers (Fig. 4F). H3K27ac ChIP-qPCR was used to evaluate the effect of CRISPRi perturbation of *CCND2* enhancers. CRISPRi significantly reduced the H3K27ac signal at the targeted site ($P < 0.001$) and also at the *CCND2* enhancer and promoter (E5) (Fig. 4G). All sgRNA targeting the enhancers significantly reduced *CCND2* expression ($P < 0.05$) (Fig. 4H). These data indicated that these EBNALP enhancers are functionally linked to the gene and are significant in activating *CCND2* expression.

## YY1 CRSIPR depletion reduces EBNALP-dependent looping

To further confirm YY1's roles in EBNALP-induced looping, YY1 was depleted with CRISPR in LCLs. Four days after puromycin selection, ~50% of YY1 was depleted while cells were mostly viable (Fig. EV2E,F). H3K27ac HiChIP was done in LCLs with YY1 depletion or control. We focused on the loops that lost in EBNALP-null virus-infected cells as YY1 is a DNA-binding protein and binds DNA independent of EBNALP in most cases. ~13% of the loops also significantly reduced upon YY1 depletion. These genes included CCND2, CD58, MIR155, and BIRC3, where EBNALP inactivation also caused the same changes (Figs. 5 and EV3). Most of the EBNALP-induced loops did not alter significantly in YY1 depletion, possibly because other mechanisms may also be involved in EBNALP-induced DNA interactions or a more complete YY1 depletion is required to see the difference.

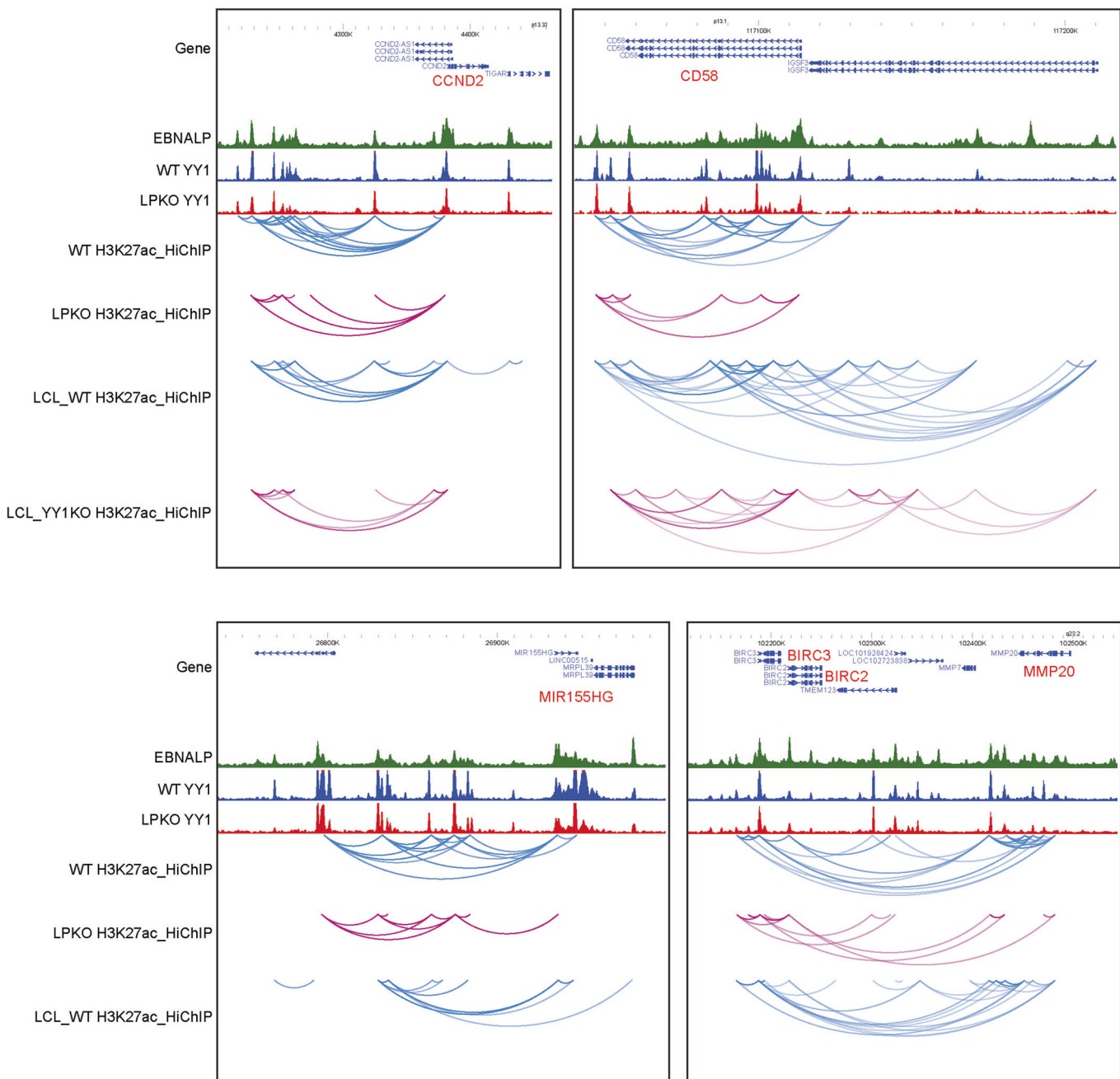

**Figure 5. YY1 depletion reduces looping at CCND2, CD58, MIR155, and BIRC2.**

YY1 was depleted using CRISPR. H3K27ac HiChIP was used to determine the looping at these loci in YY1 depletion or control cells. H3K27ac HiChIP from WT and YY1KO LCLs were performed in $n = 2$ biological replicates.

## EBNALP suppresses gene expression and enhancer looping

NBLs infected with LPKO EBV had higher expression levels of *CASP1*, *CXCR5*, and 375 other genes than wt EBV-infected NBLs, suggesting the expression of these genes were suppressed by EBNALP. Using HiChIP,

we found increased looping in cells infected with LPKO virus near these repressed genes, including BCL2L11 (Fig. 1D), CASP1, and CXCR5 (Fig. 6A). A cluster of YY1 sites ~150 kb downstream of the TSS of *CASP1* and ~24/40 kb upstream of the TSS of *CXCR5* gained YY1 signals in LPKO virus-infected cells (Fig. 6A). These sites looped extensively to TSS in LPKO virus-infected cells while the loops were

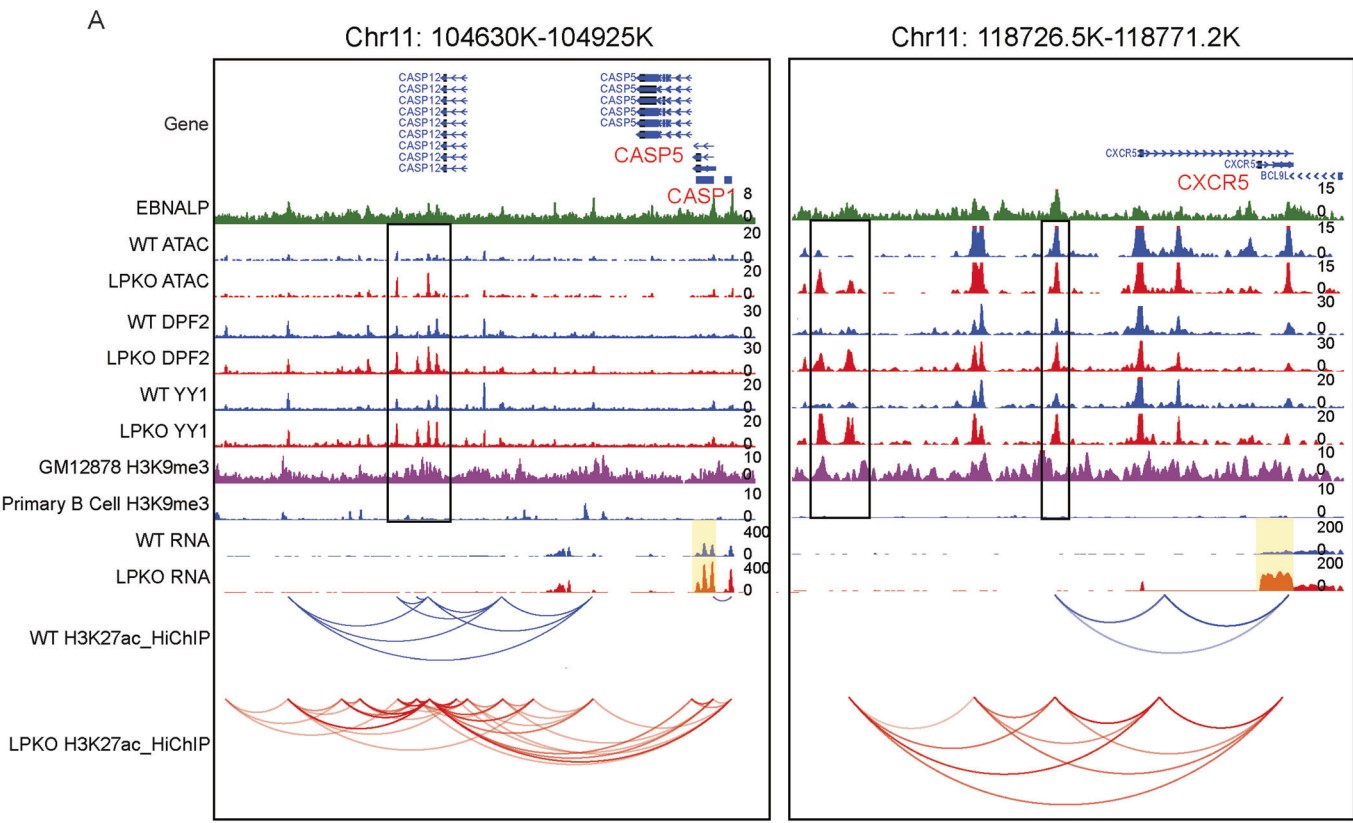

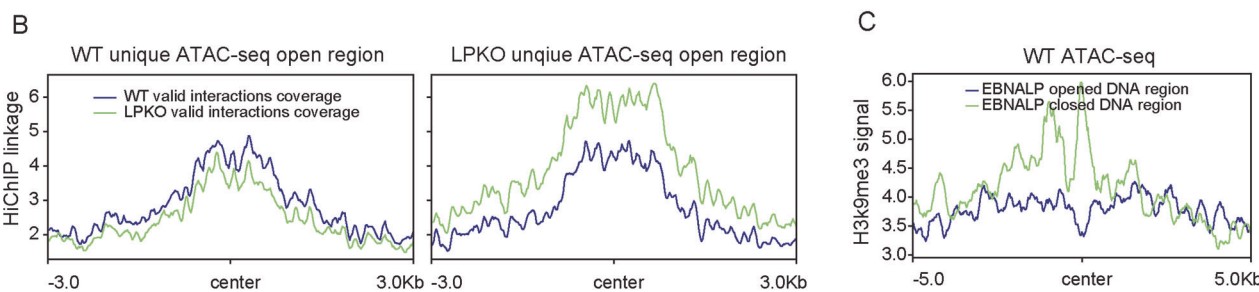

**Figure 6. EBNALP blocks YY1 DNA binding by decreasing DNA accessibility and increasing H3K9me3.**

(A) EBNALP downregulates CASP1 and CXCR5 gene expression and decreases DNA accessibility around CASP1 and CXCR5 loci. EBNALP ChIP-seq, ATAC-seq, DPF2/YY1 CUT&RUN, RNA-seq, H3K27ac HiChIP from wt and LPKO EBV-infected NBLs, LCL and primary B cells H3K9me3 ChIP-seq around CASP1 and CXCR5 loci are shown. ATAC-seq/ DPF2/YY1 CUT&RUN differential peaks are indicated in black boxes. CASP1 and CXCR5 gene bodies are highlighted in yellow. (B) LPKO unique ATAC-seq peaks has more genome interactions in LPKO EBV-infected NBLs. Average HiChIP signals (coverage) 3 kb upstream and downstream of wt (left) and LPKO (right) unique ATAC-seq peak center are shown. (C) GM12878 H3K9me3 signals (coverage) at sites where EBNALP induced open chromatin (blue line) and induced closed chromatin (green line).

much less prominent in wt virus-infected cells. The enhanced YY1 binding at this cluster correlated with the increased looping changes in LPKO virus-infected cells, suggesting that EBNALP may reduce YY1 binding to enhancer sites to reduce the enhancer looping to their target genes and downregulated the target gene expression. In addition to YY1, DPF2 DNA binding also increased in LPKO virus-infected cells (Fig. 6A).

## EBNALP decreases DNA accessibility and increases H3K9me3 marks to prevent YY1 DNA binding

To investigate the mechanisms through which EBNALP decreases YY1 DNA binding, the effect of EBNALP inactivation on chromatin openness was evaluated using ATAC-seq. NBLs infected with wt or LPKO EBV for 2 days were used in the analyses. EBNALP inactivation

decreased 361 open chromatin sites and increased 336 open chromatin sites, including *CASP1* and *CXCR5* (Figs. 6A and EV4A). LPKO EBV-infected NBL also had higher CASP1 proteins expression than wt EBV-infected NBLs (Fig. EV4B). The DNA accessibility were significantly increased in LPKO EBV-infected cells at the cluster of sites ~150 kb downstream of the *CASP1* gene where more YY1, DPF2 binding was observed. Similarly, the accessibility of *CXCR5* sites was also increased which allows DNA-binding proteins including YY1 to target these sites and to induce loop formation between enhancers and promoters. Genome-wide, at altered open chromatin sites unique for wt EBV-infected cells, more HiChIP links were observed in wt EBV-infected cells than in LPKO EBV-infected cells (Fig. 6B, left). At altered open chromatin sites unique for LPKO EBV-infected cells, more HiChIP links were observed in LPKO virus-infected cells than in wt EBV-infected cells (Fig. 6B, right). Motif analysis of EBNALP-induced accessible or compact inaccessible chromatin sites found different enriched motifs (Heinz et al, 2010). EBNALP-induced open chromatin site was enriched for NF-κB motifs while EBNALP repressed sites were enriched with RBPJ motifs (Fig. EV4C). RBPJ mostly tethers EBNA2 to DNA (Zhao et al, 2011), but ~65% of RBPJ sites that lack EBNA2 can function as transcription repressors in the absence of EBNA2 or activated Notch (Hsieh and Hayward, 1995).

Histone modification can affect chromatin accessibility. ENCODE histone modification ChIP-seq data were analyzed for sites with EBNALP-mediated reduced accessibility (EBNALP repressed). H3K9me3 ChIP-seq signals were much higher at these EBNALP repressed sites (Fig. 6A). Genome-wide, DNA sites with closed chromatin in wt EBV-infected cells had much higher H3K9me3 signals than accessible sites (Fig. 6C).

# Discussion

Looping factors are frequently mutated in various cancers (Katainen et al, 2015; Kon et al, 2013). Mutations in cohesin subunits RAD21, SMC1, SMC3, and STAG2 are seen in several myeloid neoplasms. Expression of wild-type cohesin subunits in cell lines harboring mutations suppressed the growth of these cell lines (Kon et al, 2013). STAG2 knockout caused altered enhancer–promoter interactions in Ewing sarcoma cells (Adane et al, 2021). CTCF site methylation disrupts CTCF DNA binding at contact domain boundary that divides the genome into different contact domains and leads to oncogene expression (Flavahan et al, 2016). Here, we report a viral oncoprotein that manipulates looping factor YY1 DNA binding to alter host genome organization to alter the expression of genes essential for cell cycle progression and cell death.

YY1 was cloned as a transcription repressor that binds to adeno-associated virus P5 promoter (Shi et al, 1991). It was recently reported that EBNALP can bind to YY1 through EBNALP leucine-rich motifs and the EBNALP leucine-rich motifs are required for primary B-cell transformation (Cable et al, 2024). YY1 can also activate human T lymphotropic virus type 1 LTR-driven gene expression (Wang and Goff, 2020). YY1 has been shown to regulate the viral gene expression in HPV, CMV, and EBV-infected cells (Bauknecht et al, 1992; Bauknecht et al, 1996; Brown et al, 2015; Zalani et al, 1997). YY1 interacts with retroviral integrases and facilitates moloney murine leukemia virus cDNA integration (Inayoshi et al, 2010). CTCF also regulates CMV latency by regulating the chromatin looping (Groves

et al, 2024; Groves and O'Connor, 2024). CTCF can regulate HSV latency gene expression (Lee et al, 2018; Washington et al, 2018). Together with CTCF, YY1 represses E6 and E7 expression in undifferentiated basal epithelial keratinocytes through HPV genome looping (Pentland et al, 2018). YY1 is also important for HBV cccDNA interaction with host chromosome and HBV integration (Hayashi et al, 2000; Shen et al, 2020). Our finding identified a novel mechanism through which EBV manipulates YY1 DNA binding to regulate enhancer–promoter interaction and gene expression.

YY1 is important for B-cell development. In YY1 knockout mice, somatic rearrangement in the immunoglobulin heavy-chain (IgH) locus is defective, suggesting an important role in V(D)J recombination (Liu et al, 2007). Control of 3D genome organization by YY1 through chromatin loop extrusion allows efficient V(D)J recombination (Zhang et al, 2019). YY1 dimerization facilitates the interaction between enhancers and promoters bound by YY1 (Weintraub et al, 2017). CRISPR perturbation of YY1 binding sites impairs enhancer–promoter interactions and remote enhancer-controlled gene expression (Weintraub et al, 2017). Genetic mutations of YY1 cause intellectual disability syndrome. LCLs from these patients have reduced CTCF binding and loss of H3K27ac (Gabriele et al, 2017). YY1 is also often overexpressed in various cancers, and high YY1 expression is correlated with poor prognosis (Khachigian, 2018). YY1 can induce TP53 ubiquitination and degradation (Sui et al, 2004).

YY1 DNA binding can be affected by DNA methylation. TET causes DNA demethylation. Inactivation of TET results in the disruption of YY1 DNA binding and affects long-range chromatin interactions (Fang et al, 2019). EBNALP can modulate YY1 DNA binding by direct YY1 recruitment to the EBNALP binding site or evicting YY1 from DNA by decreasing chromatin accessibility. Altered YY1 chromatin binding led to significant changes in enhancer–promoter loops and target gene expression.

We provided evidence that EBNALP can manipulate YY1 DNA binding and altered YY1 binding correlated with changes in enhancer–promoter interactions. YY1 DNA binding is signicantly reduced at YY1 and EBNALP colocalized sites in the absence of EBNALP at these sites. Given that EBNALP strongly interacts with YY1 in LCLs, it is likely that EBNALP can recruit YY1 to these sites to facilitate the enhancer–promoter interactions. YY1 partial depletion recapitulates EBNALP knockout in looping at many of these sites. It is difficult to achieve complete YY1 depletion and maintain LCL growth. Thus, we only achieved a partial effect. The partial effect can also be caused by looping factors at these sites that are independent on EBNALP and YY1. Importantly, this EBNALP-mediated alteration upregulates the expression of CCND2 which is essential for LCL growth and survival (Ma et al, 2017), thus overcoming a cell cycle checkpoint and enabling EBV transformation of NBLs (Fig. 7).

DNA accessibility determines TF DNA binding. In EBNALP knockout virus-infected NBLs, sites with increased looping had significantly higher ATAC-seq signals, allowing more YY1 DNA binding. Chromatin openness can be affected by DNA methylation and histone modifications. HDAC inhibitor can increase the DNA accessibility and YY1 DNA binding (Cusack et al, 2020). Since EBNALP associates with HDAC4 (Portal et al, 2011), it is possible that EBNALP recruits HDAC4 to reduce histone acetylation and alters the chromatin accessibility to prevent YY1 DNA binding, reducing enhancer looping to their direct target genes.

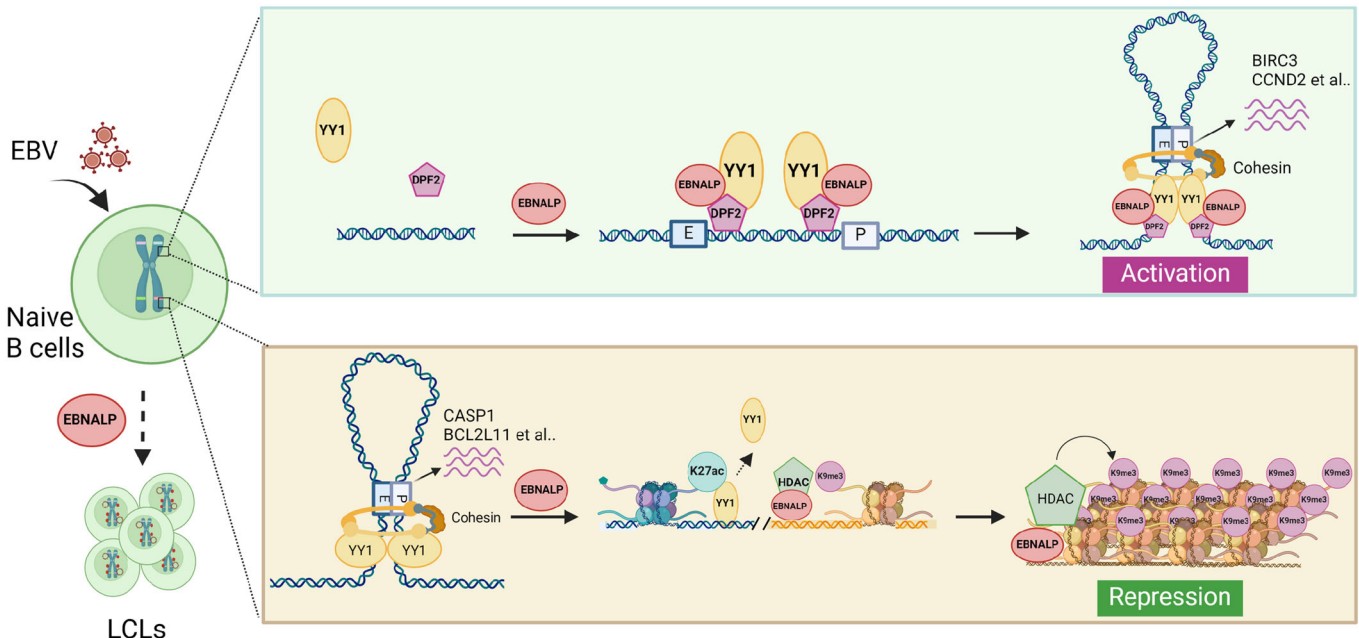

**Figure 7. Model of EBNALP-induced, YY1-mediated genome reorganization.**

Models depicting EBNALP-induced and repressed enhancer–promoter interactions. Top: DPF2 and EBNALP recruit looping factor YY1 to EBNALP-activated enhancers to increase the enhancer–promoter interaction to activate gene expression. Bottom: EBNALP recruits transcription repressors and increases H3K9me3 to decrease DNA accessibility, limiting YY1 enhancer binding and enhancer–promoter interaction to repress gene expression.

Viral infection causes ~15% of human cancers (zur Hausen, 1991). Different viruses use different mechanisms to transform normal cells into cancer cells. Small DNA tumor viruses frequently inactivate major tumor suppressor pathways, including RB and TP53 (Munger and Howley, 2002). EBV does not directly affect these tumor suppressor pathways. Instead, EBV reprograms host nuclear architecture to regulate oncogene expression by hijacking other host nuclear proteins. EBNA2 usurps the Notch pathway component RBPJ to activate the expression of viral and host genes (Henkel et al, 1994). EBNA2 enables the assembly of enhancers 400–500 kb upstream of MYC TSS and these enhancers loop to MYC TSS in an EBNA2-dependent manner through an unknown mechanism (Jiang et al, 2017). EBNA3A/C enables LCLs to overcome p16$^{INK4A}$ mediated senescence by recruiting repressors to this locus (Jiang et al, 2014; Ohashi et al, 2021; Skalska et al, 2010; Skalska et al, 2013).

EBV evolved an elegant strategy to hijack YY1 to drive cell cycle progression while also preventing apoptosis by two distinct mechanisms. On the one hand, EBNALP uses YY1 to increase enhancer–promoter interactions to activate the expression of cell cycle progression genes. On the other hand, EBNALP restricts YY1-mediated enhancer–promoter interactions to suppress proapoptotic gene expression. These findings highlight EBNALP-YY1 interaction as a novel therapeutic target in the prevention and treatment of EBV-associated lymphoproliferative diseases. Since EBNALP interaction with HDACs can suppress proapoptotic gene expression, it is possible to target this pathway to induce cancer cell death using HDAC inhibitors (Qu et al, 2017).

Incorporating multiplexed approaches including ChIP-seq for all the essential EBV genes, HiChIP and ChIA-Pet linking all the

EBV enhancers to their direct target genes, ATAC-seq identifying all EBV induced accessibility changes, RNA-seq, and genome-wide CRISPR screen for genes essential for LCL growth and survival, we now have a better understanding on how EBV contributes to B-cell oncogenesis.

## Methods

**Reagents and tools table**

| Reagent/resource | Reference or source | Identifier or catalog number |
|---|---|---|
| **Antibodies** | | |
| YY1 antibody | Abcam | ab109237, RRID:AB_10890662 |
| DPF2 antibody | Thermo Fisher Scientific | A303-596A, RRID:AB_11125151 |
| H3K27ac antibody | Abcam | ab4729, RRID:AB_2118291 |
| Mouse monoclonal anti-EBNALP antibody [JF186] | Dr. Elliott Kieff | N/A |
| Mouse monoclonal anti-EBNALP antibody [4D3] | Dr. Elliott Kieff | N/A |
| rH2AX antibody | Millipore sigma | 05-636, RRID:AB_309864 |
| CASP1 antibody | Cell Signaling Technology | 2225S, RRID:AB_2243894 |
| GAPDH antibody | Cell Signaling Technology | 5174S, RRID:AB_10622025 |
| HA95 antibody | Novus biologicals | H00010270-M01, RRID:AB_489990 |
| TP53BP1 antibody | Novus biologicals | NB100-304, RRID:AB_10003037 |

| Reagent/resource | Reference or source | Identifier or catalog number |
|---|---|---|
| TOP2B antibody | Millipore sigma | HPA024120, RRID:AB_1858191 |
| Anti-mouse IgG | Santa Cruz Biotechnology | sc-2025, RRID:AB_737182 |
| Anti-rabbit IgG | Cell Signaling Technology | 2729, RRID:AB_1031062 |
| Anti-rabbit IgG, HRP-linked Antibody | Cell Signaling Technology | Cat#7074; RRID: AB_2099233 |
| Anti-mouse IgG, HRP-linked Antibody | Cell Signaling Technology | Cat#7076; RRID: AB_330924 |
| Mouse monoclonal anti-EBNA2 antibody [PE2] | Dr. Elliott Kieff | N/A |
| **Bacterial and virus strains** | | |
| WT B95.8 Epstein–Barr Virus | Dr. Wolfgang Hammerschmidt | N/A |
| EBNALP KO B95.8 Epstein–Barr Virus | Dr. Wolfgang Hammerschmidt | N/A |
| **Chemicals** | | |
| TransIT-LT1 transfection reagent | Mirus Bio | MIR 2306 |
| Puromycin dihydrochloride | FISHER SCIENTIFIC | A1113803 |
| Blasticidin | InvivoGen | ant-bl-5 |
| BSA | NEB | B9000S |
| Penicillin-Streptomycin | Thermo Fisher Scientific | 15070063 |
| Agencourt AMPure XP | Beckman Coulter | A63881 |
| Biotin-14-dATP | Thermo Fisher Scientific | 19524016 |
| MboI | NEB | R0147M |
| Dynabeads® MyOne™ Streptavidin C-1 | Thermo Fisher Scientific | 65001 |
| Dynabeads® Protein A for Immunoprecipitation | Thermo Fisher Scientific | 10002D |
| RPMI | FISHER SCIENTIFIC | 11875085 |
| FBS | Thermo Fisher Scientific | 16000044 |
| Protease inhibitors | Roche | 11697498001 |
| DNA Polymerase I, Large (Klenow) Fragment | NEB | M0210 |
| 10X NEB T4 DNA ligase buffer with 10 mM ATP | NEB | B0202 |
| T4 DNA Ligase | NEB | M0202 |
| Proteinase K | NEB | P8107S |
| ProLong™ Gold Antifade Mountant with DAPI | Thermo Fisher | P36935 |
| Phusion® High-Fidelity PCR Master Mix with HF Buffer | NEB | M0531S |
| **Critical commercial assays** | | |
| iScript Supermix cDNA synthesis kit | Bio-Rad | 1708841 |
| CUTANA™ ChIC/ CUT&RUN Kit | Epicypher | 14-1048 |
| Illumina DNA library prep kit | NEB | E7645S |

| Reagent/resource | Reference or source | Identifier or catalog number |
|---|---|---|
| NEBNext® Ultra™ II Directional RNA Library Prep with Sample Purification Beads | NEB | E7765L |
| RNeasy Mini kit | Qiagen | 74004 |
| RosetteSep Human B Cell Enrichment Cocktail and | STEMCELL | 15064 |
| EasySep Human naive B Cell Enrichment Kits | STEMCELL | 17254 |
| Immobilon Forte Western HRP substrate | Millipore Sigma | WBLUF0500 |
| CellTiter-Glo Luminescent Cell Viability Assay | Promega | G7570 |
| Illumina Tagment DNA Enzyme and Buffer Small Kit | Illumina | 20034197 |
| MycoAlert Mycoplasma Detection Kit | Lonza | Cat# LT07-218 |
| DNA Clean & Concentrator-5 Kit, Capped columns, 50 Preps/Unit | GENESEE SCIENTIFIC CORP | 11-302 C |
| **Experimental models: cell lines** | | |
| GM12878 lymphoblastoid cell line (LCL) | Coriell Institute for Medical Research | N/A |
| GM12878-Cas9 | This paper | N/A |
| 293T | ATCC | CRL-3216 |
| GM12878-dCas9 | This paper | N/A |
| IB4-Cas9 | This paper | N/A |
| 293_WTEBV | Dr.Wolfgang Hammerschmidt | N/A |
| 293_LPKOEBV | Dr.Wolfgang Hammerschmidt | |
| EBV+ Burkitt lymphoma Daudi cell line | ATCC | CCL-213 |
| Naïve B cells | This paper | N/A |
| **Recombinant DNA** | | |
| pLenti-Guide-puro | Addgene | 52963 |
| lenti_dCas9-KRAB-MeCP2 | Addgene | 122205 |
| pLentiCas9-Blast | Addgene | 52962 |
| pLenti-Guide-puro-DPF2sg | This paper | N/A |
| pLenti-Guide-puro-CCND2E1isg1 | This paper | N/A |
| pLenti-Guide-puro-CCND2E1isg2 | This paper | N/A |
| pLenti-Guide-puro-CCND2E2isg1 | This paper | N/A |
| pLenti-Guide-puro-CCND2E2isg2 | This paper | N/A |
| **Software and algorithms** | | |
| FeatureCounts v1.6.1 | | http://subread.sourceforge.net/ |
| EdgeR v3.33.1 | | https://bioconductor.org/packages/release/bioc/html/edgeR.html |
| DESeq2 v1.14.1 | Love et al, 2014 | https://bioconductor.org/packages/release/bioc/html/DESeq2.html |

| Reagent/resource | Reference or source | Identifier or catalog number |
|---|---|---|
| FastQC v0.11.3 | | https://www.bioinformatics.babraham.ac.uk/projects/fastqc |
| STAR v2.7.3a | | https://github.com/alexdobin/STAR |
| EnhancedVolcano v1.7.14 | | https://bioconductor.org/packages/EnhancedVolcano |
| Bowtie2 v2.4.1 | | https://github.com/BenLangmead/bowtie2 |
| Picard v.2.24.2 | | http://broadinstitute.github.io/picard/ |
| deepTools v3.5.0 | | https://github.com/deeptools/deepTools |
| HiC-Pro v2.11.4 | | https://github.com/nservant/HiC-Pro |
| hichipper v0.7.7 | | https://github.com/aryeelab/hichipper |
| diffloop v1.17.0 | | https://github.com/aryeelab/diffloop |

### Induction and purification of WT and EBNALP_KO EBV

EBNALP_KO EBV was made by introducing the stop codon into each EBNALP W repeat and assembled into EBNALP mutant. The EBNALP mutant was then used to replace the wt EBNALP in wt EBV BAC (Pich et al, 2019). 293 cells containing WT EBV (293_WT) or EBNALP_KO EBV (293_LPKO) were passaged 24 h before transfection. Transfection was performed when cells reached 60–70% confluence. Cells were transfected with 10 μg pcDNA3.1-BZLF1 and pcDNA3.1-BALF4 pre-mixed with TransIT®-LT1 transfection reagent (Mirus, Cat: MIR 2306) according to the manufacturer's instructions. Fresh media were changed the next day. Supernatants containing released WT EBV or EBNALP_KO EBV were collected 72 h later and filtered through a 0.45-μm filter followed by ultracentrifuge at 25,000 rpm for 2 h. Following centrifugation, viral pellets were resuspended with fresh RPMI medium without serum and stored at 4 °C.

### Naive B-cell purification

De-identified blood cells were purchased from Gulf Coast Regional Blood Center, Huston USA, following institutional guidelines. The Epstein–Barr virus studies described in this paper were approved by the Brigham & Women's Hospital Institutional Review Board. B cells were purified via negative selection with RosetteSep Human B Cell Enrichment Cocktail and EasySep Human naive B Cell Enrichment Kits (StemCell Technologies), according to the manufacturer's protocols.

### Determination of virus titer

WT and EBNALP_KO EBV titers were determined using Daudi cells. Harvested viruses were resuspended in RPMI 1640 medium. Different amounts of virus suspension (10 μl, 100 μl, and 1 ml) were incubated with Daudi cells at $2.5 \times 10^5$/ml. The percentage of Daudi cells infected by WT or EBNALP_KO EBV was determined at using FACS (GFP positive) 72 h post infection. The amounts of virus that infected 50% of the Daudi were used to infect naive B cells.

### HiChIP

H3K27ac HiChIPs were performed as previously described (Wang and Goff, 2020). Briefly, 10 million naive B cells were collected 2 days after WT or EBNALP_KO EBV infection and crosslinked with 1% formaldehyde. Chromatin was then digested using MboI restriction enzyme (New England Biolabs), and the DNA ends were filled in with Biotin-14-dATP (Thermo Fisher) and other nucleotides. After ligation with T4 ligase, DNA was sonicated into small fragments and diluted 10-fold with HiChIP dilution buffer, followed by incubation with H3K27ac antibody at 4 °C overnight. The next day, chromatin–antibody complexes were captured by Dynabead Protein A. DNA tagged with Biotin-14-dATP was further enriched with Streptavidin C-1 beads (Thermo Fisher). Libraries were generated using Tn5 followed by PCR amplification. HiChIP samples were two-size selected with AMPure XP Beads and sequenced on the Illumina NextSeq 500 platform.

### CUT &RUN

CUT&RUN was done following the protocol from CUTANA™ ChIC/CUT&RUN Kit (Epicypher, 14-1048). In brief, 0.5 million cells per sample were collected followed by nuclei isolation with nuclei extraction buffer. Extracted nuclei were captured with activated ConA beads. In all, 1 μg antibody against the protein of interest was added to the nucleus solution and incubated at 4 °C with shaking overnight. pAG-MNase was then used to capture the antibody, followed by cleaving the DNA nearby. Cleaved DNA released into the solution was purified for library preparation. The DNA library was prepared using the Illumina DNA library prep kit (E7645S), and sequenced on the Illumina NextSeq 500 platform.

### ATAC-seq

ATAC-seq was done with 50,000 cells. After preparation of nuclei, the nuclei were mixed with transposase reaction. The DNA was then purified and PCR amplified, and sequenced on a NextSeq 500.

### RNA-seq

Total RNAs were extracted using RNeasy Mini kit (Qiagen) according to the manufacturer's instructions. 400 ng extracted RNAs were used for library preparation. Poly(A)-tagged RNAs were enriched using a NEBNext Poly(A) mRNA Magnetic Isolation Module (New England BioLabs). RNA-seq libraries were prepared using the NEBNext Ultra II Directional RNA Library Prep kit (New England BioLabs) according to the manufacturer's instructions. RNA-seq libraries were sequenced on the Illumina NextSeq 500 platform.

### CRISPR-cas9 knockout

IB4 and GM12878 cells were transduced with CAS9 expressing lentiviruses followed by 5 μg/ml blasticidin selection for 7 days to eliminate uninfected cells. The expression of CAS9 was validated by western blot. sgRNAs targeting genes of interest were designed with online tools benching (benchling.com) and cloned into pLenti-Guide-puro vector according to protocols from Dr. Zhang Feng's lab (https://zlab.bio/). Lentiviruses were prepared by transfecting HEK293T cells

with pCMV-VSVG (# 8454; Addgene), psPAX2 (#12260; Addgene), and pLenti-Guide-puro plasmid expressing gRNA with TransIT-LT1 transfection reagent (Mirus) according to the manufacturer's instructions. Forty-eight hours later, lentiviruses were harvested and used to infect target cells in which CAS9 was stably expressed. Forty-eight hours after infection, cells were selected with 3 µg/mL puromycin and allowed to outgrowth for another 3 days before testing gene knockout efficiency by western blot.

## CRISPRi repression

Plasmid dCas9-KRAB-MeCP2 (#110821) purchased from addgene was packaged into lentiviruses and then used to infect GM12878 cells, followed by selecting with 5 µg/ml blasticidin for 7 days to eliminate uninfected cells. GM12878 cells expressing fused protein dCas9-KRAB-MeCP2 were then infected with lentiviruses expressing sgRNAs targeting loci of interest. Cells were then selected with 3 µg/mL puromycin for another 3 days to eliminate uninfected cells. The efficiency of CRISPRi was tested by ChIP-qPCR or qRT-PCR.

## qRT-PCR

Total RNAs were extracted using RNeasy Mini kit (Qiagen) and cDNAs were generated using iScript reverse transcription supermix kit (Bio-Rad). Power SYBR green master mix (Thermo Fisher) was used to quantitate the mRNA level. Data were normalized to endogenous control GAPDH (glyceraldehyde-3-phosphate dehydrogenase). Relative expression was calculated using $2\text{-}\Delta\Delta Ct$ method, with the normalized Ct value of the untreated or mock-treated sample being the baseline. At least three independent experiments were performed.

## ChIP-qPCR

One million cells were harvested and fixed with 1% formaldehyde. Cells were lysed with cell lysis buffer, and DNA was sonicated into fragments ranging from 200 to 1000 bp with bioruptor (Diagenode). Sonicated chromatin was 10-fold diluted with ChIP dilution buffer, pre-cleared with protein A agarose beads, followed by incubation with 4 µg H3K27ac or IgG antibody with rotating at 4 °C overnight. The next day antibody-chromatin complexes were precipitated with 60 µl Protein A agarose/Salmon Sperm DNA beads. After precipitation, beads were washed extensively with low salt, high salt, and licl wash buffer. DNA was then eluted and reverse crosslinked with NaCl and Proteinase K. DNA was purified using QIAquick Spin columns (Qiagen). qPCR was used to quantify the chipped DNA. The amount of chipped DNA was normalized to input DNA.

## Cell growth assay

Cell growth was measured with kit CellTiter-Glo® Luminescent Cell Viability Assay (Promega) according to the manufacturer's instructions. The value from new purified naive B cells was set to 1.

## Immunofluorescence analysis

Cells were harvested and washed with PBS once. 20 µl cells were seeded onto the slides and allowed to dry by incubating slides in

37 °C for 1 to 2 h. Cells were then fixed with 10 µL 3% PFA at room temperature for 15 min. After fixation, cells were washed with PBS twice and permeabilized with 10 µL of 0.2% TritonX. 3% BSA was used for blocking, followed by incubation with primary antibodies (1:100 dilution) at 4 °C overnight. The next day, secondary antibodies (1:500) were then added and incubated at room temperature for 30 min. Finally, cells were washed twice with PBS and were stained overnight with ProLong™ Gold Antifade Mountant with DAPI.

## Co-immunoprecipitation

Cells were harvested and washed with PBS twice. For EBNALP and YY1 co-immunoprecipitation, 20 million cells were lysed in 1 ml cell lysis buffer by incubation at 4 °C for 30 min, followed by centrifugation at 13,000 rpm for 10 min. Supernatants were then collected and pre-cleared with 4 µg IgG and 20 µl protein A/G beads with rotating at 4 °C for 1 h. Protein A/G beads and IgG antibodies were then removed by placing tubes to a magnetic stand. Pre-cleared supernatants were then collected, and 4 µg EBNALP (4D3) or YY1 antibody was added to capture EBNALP or YY1 by rotating at 4 °C overnight. IgG was used as a negative control. The next day, 20 µl protein A/G beads were used to precipitate antibody and protein complexes. The interactions of EBNALP and YY1 were then detected by western blot. Different from EBNALP and YY1 co-immunoprecipitation, 50 million cells were used for EBNALP and DPF2 co-immunoprecipitation. Overall, 50 million IB4 cells were lysed with 1 ml cell lysis buffer for 10 min at 4 °C followed by sonicating once for 3 s and then lysed again in 4 °C for another 20 min.

## Sequencing

Illumina HiSeq was used to sequence the samples using paried-end reads, 35 bp. The sequencing depth were between 10 and 30 million reads.

## Sequencing quality evaluation

Sequencing adapters were trimmed at the core facility after sequencing. All sequencing reads, including RNA-seq, CUT&RUN, ATAC-seq and HiChIP, were quality controlled using FastQC v0.11.3 to ensure no significant GC bias and PCR artifacts (https://www.bioinformatics.babraham.ac.uk/projects/fastqc).

## RNA-seq data processing

Paired-end RNA-seq reads of WT and LPKO samples were aligned to human (hg19) genome using STAR v2.7.3a with the aligning parameters "--outSAMprimaryFlag AllBestScore". featureCounts v1.6.1 was used to calculate gene expression reads counts based on GENCODE v34 annotation, with the parameters "-p -s 0 -t exon -g gene_id -Q 10 –ignoreDup". DESeq2 v1.29.7 was then used to normalize gene read counts and estimate differentially expressed genes. Differentially expressed genes were selected based on fold change ($> 2$), FDR ($< 0.05$). Volcano plot was generated by using EnhancedVolcano v1.7.14 (https://bioconductor.org/packages/EnhancedVolcano).

## Pathway analysis

ShinnyGO 0.77 was used for the pathway analysis. The link is: http://bioinformatics.sdstate.edu/go/.

## CUT&RUN, ChIP-seq, and ATAC-seq data processing

CUT&RUN reads for TOP2B, DPF2 and YY1 were aligned to human (hg19) genome using Bowtie2 v2.4.1 with the parameters "-I 10 -X 700 --local --very-sensitive-local --no-discordant --no-mixed --no-unal --phred33 -k 1". PCR duplicated reads were marked and removed using Picard (http://broadinstitute.github.io/picard/) and Samtools. Protein binding peaks were called using MACS v2.2.7.1 on each sequencing sample with the parameters "--nomodel -f BAMPE ". Peaks located in blacklist regions were removed in downstream analysis. Peaks were further merged across samples to create a unified peak list for each protein. CUT&RUN reads located in the unified peaks were quantified using featureCounts with MAPQ > 10, followed by differential binding testing using edgeR v3.33.1 for each protein, similar to the strategy proposed by Ross-Innes et al (Ross-Innes et al, 2012). Significantly differential binding peaks were selected based on fold change (> 1.5), FDR (< 0.05) and mean normalized CPM (counts per million) (> 3).

ChIP-seq reads were aligned to human (hg19) and EBV (Akata) genomes using Bowtie2 v2.4.1 under default settings except parameter -k was set to 1. ChIP-seq peaks were called using MACS v2.2.7.1 with the parameters "--nomodel -f BAMPE -g hs" on each replicated sample. Peaks located in blacklist regions were removed in the downstream analysis.

ATAC-seq reads were aligned to human (hg19) and EBV (Akata) genomes using Bowtie2 v2.4.1 under default settings except parameter -k was set to 1. Peaks were called using MACS v2.2.7.1 with the parameters "--nomodel -f BAMPE -g hs --shift -37 --extsize 73" on each replicated sample. Peaks located in blacklist regions were removed in downstream analysis. The R package Rsubread v2.3.7 was used to calculate reads counts in each peak. The R package edgeR v3.33.1 was used for differential peak analysis with the standard "logFC> 0.585 & $P$.adj <0.05 & logCPM>3". Sequencing fragment size density was estimated to confirm sequencing quality based on the clear nucleosome phasing patterns.

## HiChIP data processing

HiChIP paired-end reads were mapped to human (hg19) and EBV (Akata) genomes using HiC-Pro v2.11.4 (default settings with LIGATION_SITE set as GATCGATC for MboI). Significant loops identified with hichipper v0.7.7 using the peak calling method "EACH, ALL". On average, the percentage of valid interaction read pairs are ~47%, indicating the high data quality. Significant HiChIP loops were further filtered with mango $p$ value < 0.01, followed by differential loops were detected using diffloop v1.17.0 with the standard "mango.FDR < 0.01 & abs(logFC)>1". Differential looping was selected based on FDR (< 0.01) and log change (> 2). Loops were categorized based on the locations of the two anchors overlapping with genome-wide promoters (−3kbp ~ 0 bp of transcription start sites) and enhancers (H3K27ac binding except promoters). Basically, P–P loops indicate both anchors of the loops are located at promoters, while E-E loops indicate both anchors are located at enhancers. The E–P loops denote that one anchor is located at promoters, and the other is located at enhancers.

For HiChIP normalization, replicates from each group were first merged with the software samtools v1.10. After loop calling, the filtered intra-chromosome loops ranging from 5kb- 2 M bps were selected. The intensity of loops are normalized based on sample scale factors which were estimated by edgeR v3.33.1 using the HiChIP coverage of common loop anchors between between WT and EBNALP_KO groups.

## Public ChIP-seq data

The GM12878 H3K9me3 ChIP-seq data were downloaded from GEO (GSM733664). The ChIP-seq data of Primary B Cell H3K9me3 and GM12878 DPF2 were downloaded from ENCODE (ENCFF807XIY, ENCSR509FWH). The ChIP-seq data for IB4 EBNALP were downloaded from GEO (GSE49338), followed by the same analysis procedure as CUT&RUN with the exception of gene mapping using Bowtie2 v2.4.1withparameters setting to "-k 1".

## Statistical analysis

The statistical significance of differences between means from at least three experiments was determined using unpaired Student's $t$ tests.

## AI assistance in writing

ChatGPT and Grammarly were used to spell check and proofread part of the manuscript.

# Data availability

All sequencing data generated for this manuscript have been deposited in Gene Expression Omnibus (GEO) under accession GSE277748, https://www.ncbi.nlm.nih.gov/geo/query/acc.cgi.

The source data of this paper are collected in the following database record: biostudies:S-SCDT-10_1038-S44319-024-00357-6.

# Peer review information

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

## Acknowledgements

The authors thank Dr. Elliott Kieff for his tremendous insights, for sharing his knowledge and for helpful discussions. The authors thank Dr. Yasushi Kawaguchi for anti-EBNALP 4D3 antibody. This work was supported by NIH (R01AI123420 and R01CA047006 to BZ, 4R00DE030215-03 to CW and R01AI164709 and DE033907 to BG, and R35GM155298 to MT). BZ is also supported by Fund to Sustain Research Excellence from Brigham Research Institute.

## Author contributions

**Chong Wang**: Conceptualization; Data curation; Software; Formal analysis; Supervision; Validation; Investigation; Visualization; Methodology; Writing—original draft; Writing—review and editing. **Merrin Manlong Leong**: Data curation; Investigation; Visualization. **Weiyue Ding**: Formal analysis; Methodology. **Yohei Narita**: Investigation; Visualization. **Xiang Liu**: Data curation; Software; Formal analysis. **Hongbo Wang**: Investigation. **Stefanie P T Yiu**: Resources. **Jessica Lee**: Investigation; Visualization. **Katelyn R S Zhao**: Investigation; Visualization. **Amy Cui**: Investigation; Visualization. **Benjamin Gewurz**: Resources; Supervision; Funding acquisition. **Wolfgang Hammerschmidt**: Resources. **Mingxiang Teng**: Conceptualization; Software; Formal analysis; Supervision; Funding acquisition; Investigation; Visualization; Writing—original draft; Writing—review and editing. **Bo Zhao**: Conceptualization; Resources; Formal analysis; Supervision; Funding acquisition; Writing—original draft; Writing—review and editing.

Source data underlying figure panels in this paper may have individual authorship assigned. Where available, figure panel/source data authorship is listed in the following database record: biostudies:S-SCDT-10_1038-S44319-024-00357-6.

## Disclosure and competing interests statement

The authors declare no competing interests.

# Expanded View Figures

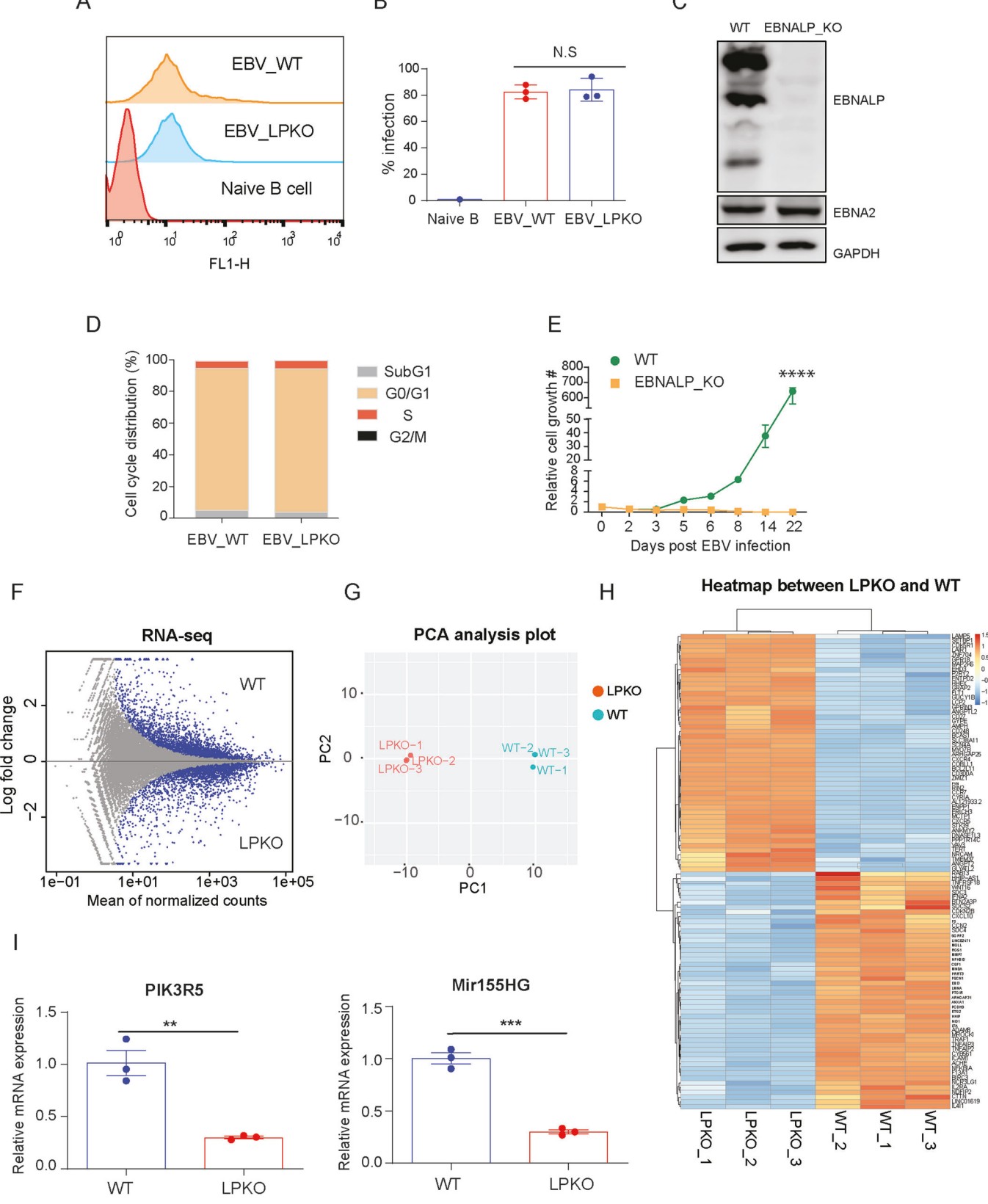

◄ **Figure EV1. Characterization of wt and LPKO EBV infected NBLs.**

(A) Flow cytometry analyses of wt and LPKO EBV-infected NBLs. Uninfected NBLs were used as a negative control. (B) Statistic results of percentage of NBLs infected with WT or LPKO EBV ($n = 3$). N.S (no significance). ($n = 3$ biological replicates). (C) Western blots detecting EBNALP and EBNA2 expression after WT or LPKO EBV infection of NBLs. GAPDH was used as a loading control. Blots are representative of $n = 2$ replicates. (D) Cell cycle analysis of NBLs infected with WT or LPKO EBV. Cell cycle is average from $n = 3$ replicates. (E) Relative cell growth curve of NBLs infected with WT or LPKO EBV. Day 0 was normalized to 1. ($n = 6$ biological replicates). Statistical significance was tested between WT and LPKO groups at day 28. ($P$ value < 0.0001 was shown as <0.0001. $P < 0.0001$). (****$P < 0.0001$). (F) Scatterplot displaying changes in gene expression (log fold change) against all the genes between WT and LPKO EBV infection of NBLs. Differential genes were shown as blue dots. Genes from WT group were plotted above, and genes from LPKO group were plotted below. (G) PCA analysis of RNA-seq three replicates from WT and LPKO EBV-infected NBLs. (H) Heatmap displaying some genes that were differentially expressed between WT and LPKO EBV infected NBLs. (I) RT-qPCR detecting PIK3R5 and Mir155HG transcription from WT and LPKO infected NBLs. RT-qPCR was performed in $n = 3$ biological replicates. Statistical significance was tested between WT and LPKO groups. (For PIK3R5, $P = 0.004$; MIR155HG, $P = 0.0002$). A two-tailed unpaired $t$ test was used for statistical analyses. The error bars indicate the SEM for the averages of $n = 3$ biological replicates. (**$P < 0.01$, ***$P < 0.001$). Source data are available online for this figure.

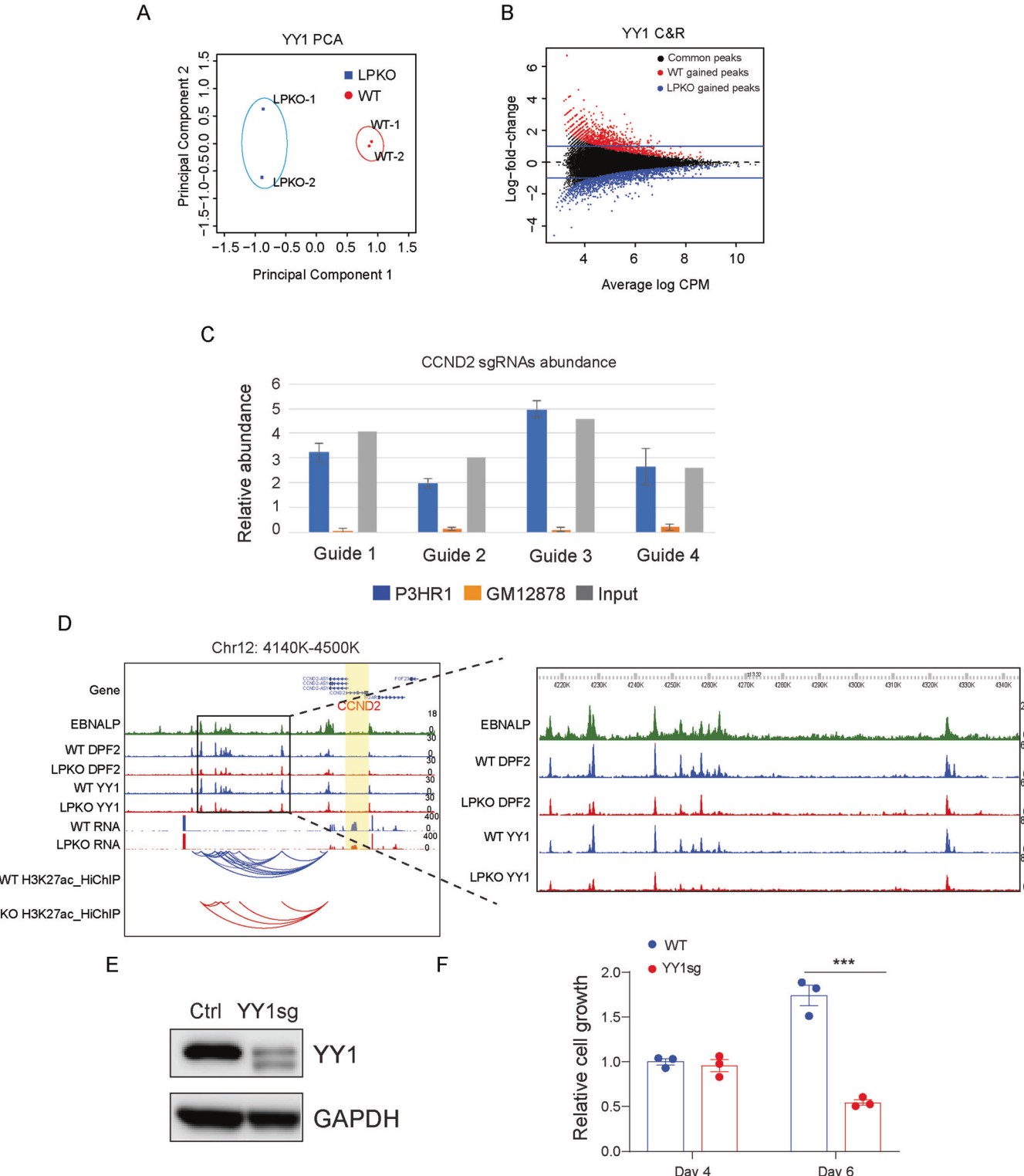

◀  **Figure EV2.  Characterization of YY1 DNA binding genome-wide, at CCND2 locus and YY1 depletion.**

(A) PCA analysis of YY1 CUT&RUN two replicates from WT and LPKO EBV-infected NBLs. (B) Scatterplot displaying YY1 DNA-binding loci changes between WT and LPKO EBV infection of NBLs. Red dots are YY1 CUT&RUN peaks uniquely detected from WT EBV-infected NBLs (EBNALP gained peaks), blue dots are YY1 CUT&RUN peaks uniquely detected from LPKO EBV-infected NBLs (EBNALP reduced peaks). Black dots within the green dot line are YY1 CUT&RUN peaks unchanged between the two groups (Stable peaks). (C) CCND2 is essential for LCL growth and survival. CCND2 CRISPR depletion prevents LCL growth. CRISPR was from $n = 2$ biological replicates. (D) Zoom in on DPF2 and YY1 CUT&RUN tracks from WT and LPKO infected NBLs at CCND2 locus. (E) Western blot displaying YY1 protein level after depletion of YY1 with CRISPR-cas9. GAPDH was used as loading control. Blots are representative of $n = 2$ replicates. (F) Relative cell growth of WT and YY1 depleted LCLs. Cell growth was monitored by CTG assay. Cell growth is from $n = 3$ replicates. Statistical significance was tested between WT and YY1sg groups. (For Day 6, $P = 0.0006$). A two-tailed unpaired $t$ test was used for statistical analyses. The error bars indicate the SEM for the averages. (***$P < 0.001$). Source data are available online for this figure.

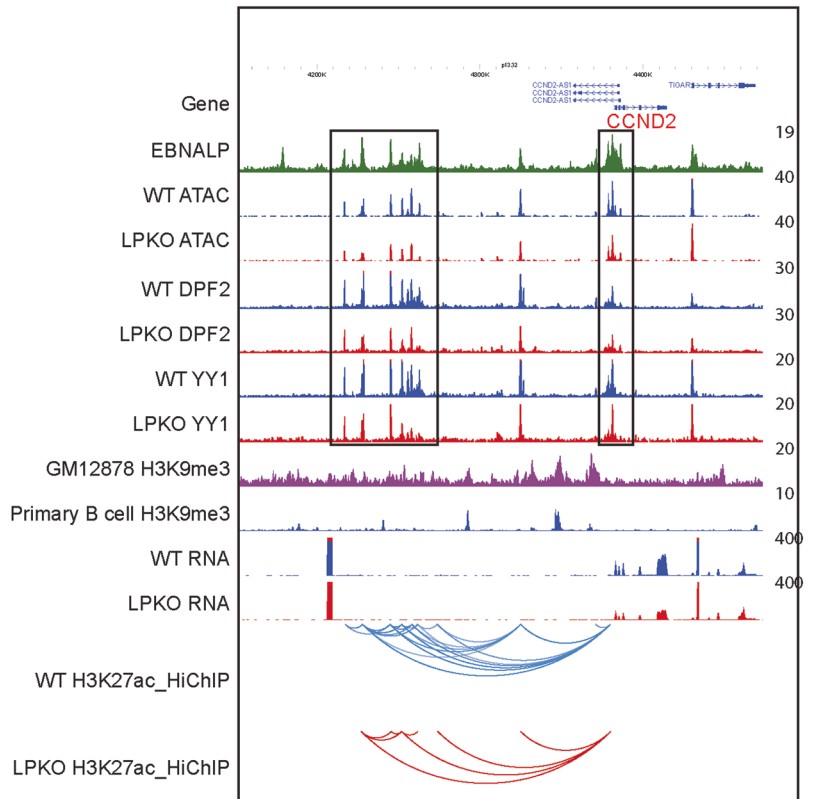
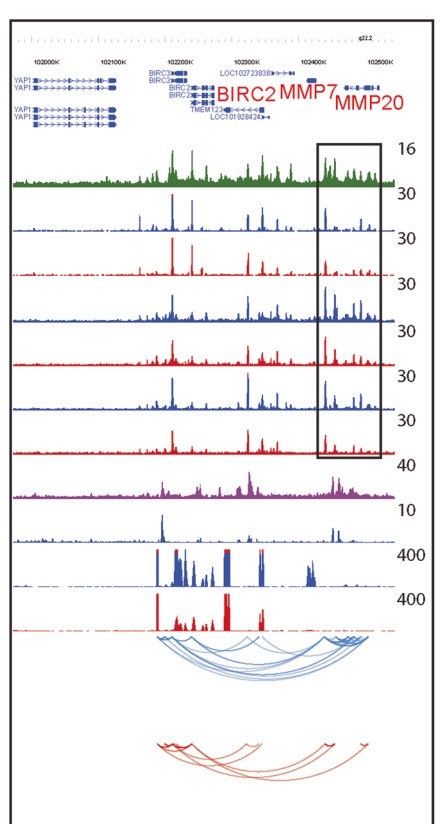

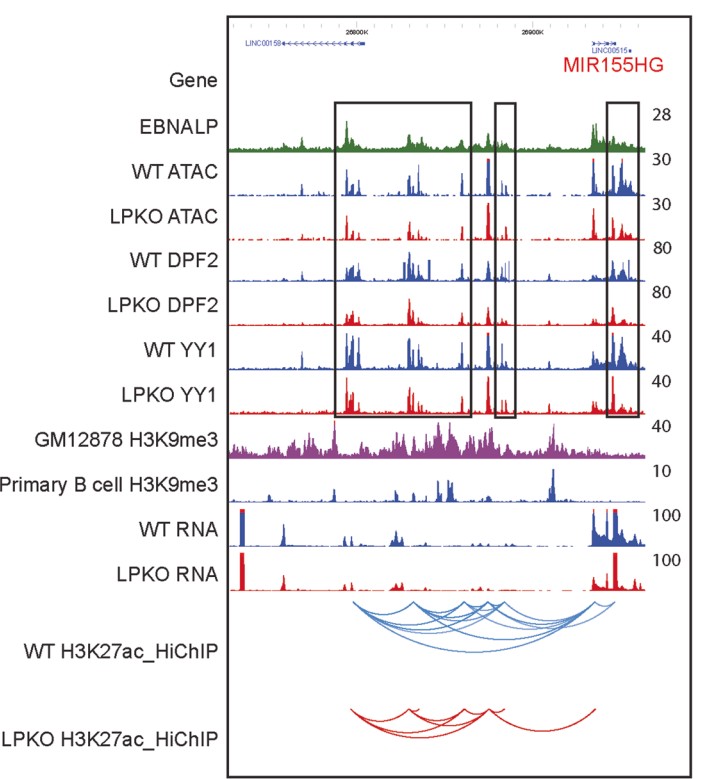
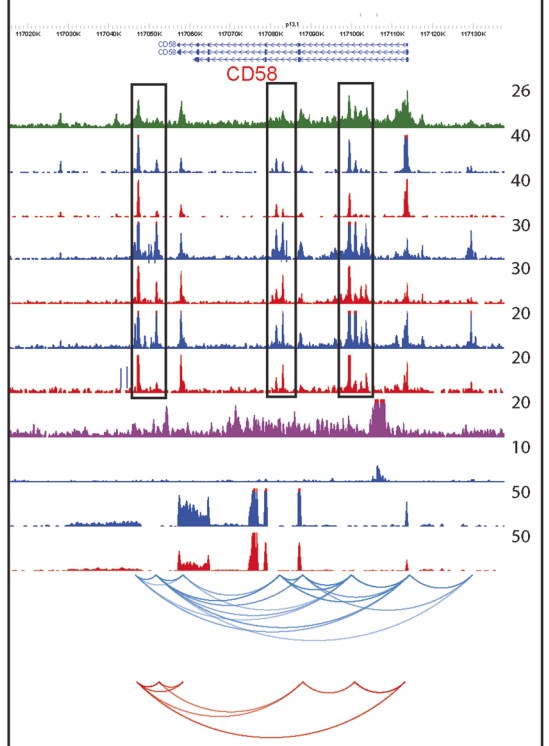

Figure EV3.  ChIP-seq, Cut&Run, ATAC-seq, RNA-seq, H3K27ac HiChIP tracks from wt or LPKO EBV-infected cells, LCL or RBL H3K9me3 and EBNALP ChIP-seq tracks at the CCND2, BIRC2, MIR155, and CD58 loci are shown.

Peak height are indicated on the right side of the corresponding tracks.

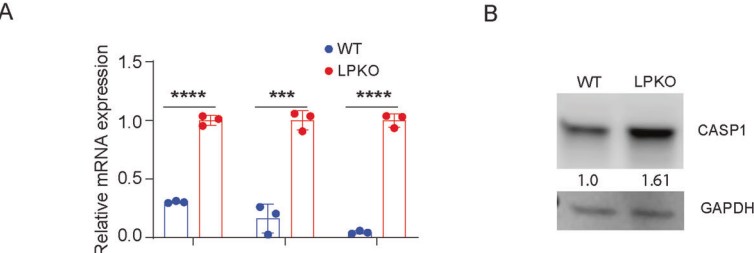

C

WT unique ATAC-seq open region motif

| TF | Motif | P-value |
|---|---|---|
| NFkB-p65 | | 1e-28 |
| EWS:ERG | | 1e-18 |
| ELF4 | | 1e-16 |
| ETV2 | | 1e-15 |
| ETS1 | | 1e-15 |
| ELF5 | | 1e-14 |
| ELF3 | | 1e-14 |
| PU.1:IRF8 | | 1e-13 |

LPKO unique ATAC-seq open region motif

| TF | Motif | P-value |
|---|---|---|
| ETV2 | | 1e-18 |
| ETS1 | | 1e-17 |
| RBPJ | | 1e-15 |
| ERG | | 1e-14 |
| ETV1 | | 1e-14 |
| RUNX1 | | 1e-11 |
| RUNX2 | | 1e-11 |
| EBF2 | | 1e-10 |

**Figure EV4. LPKO EBV induces CASP1 and enrichment of unique motifs at EBNALP regulated ATAC-seq sites.**

(A) EBNALP downregulates CASP1, CASP5 and CXCR5 transcription. CASP1, CASP5 and CXCR5 mRNA from RNA-seq. Statistical significance was tested between the WT and LPKO groups ($P$ value less than 0.0001 is shown as <0.0001. For CASP1, $P < 0.0001$; CASP5, $P = 0.0006$; CXCR5, $P < 0.0001$). (***$P < 0.001$, ****$P < 0.0001$). The error bars indicate the SD for the averages of $n = 3$ biological replicates. (B) Western blot detecting CASP1 expression after WT and LPKO EBV infection of NBLs. In the absence of EBNALP, EBV upregulated CASP1 expression, but no cleaved CASP1 protein was detected. GAPDH was used as a loading control. The numbers represent relative protein band intensity measured with Image Studio, quantified by normalizing to GAPDH. Blots are representative of $n = 2$ replicates. (C) Motif analysis of EBNALP-induced accessible or compact inaccessible chromatin sites. Source data are available online for this figure.

