## [Peer Review File · EMBO Reports]

Viral oncogene EBNA1P regulates YY1 DNA binding and alters host 3D genome organization

Chong Wang, Merrin Leong, Weiyue Ding, Yohei Narita, Xiang Liu, Hongbo Wang, Stefanie Yiu, Jessica Lee, Katelyn Zhao, Amy Cui, Benjamin Gewurz, Wolfgang Hammerschmidt, Mingxiang Teng, and Bo Zhao

Corresponding author(s): Bo Zhao (bzhao@bwh.harvard.edu), Chong Wang (wan01642@umn.edu), Mingxiang Teng (mingxiang.teng@moffitt.org)

Review Timeline:

Submission Date:	12th Jun 24
Editorial Decision:	9th Sep 24
Revision Received:	10th Sep 24
Editorial Decision:	5th Nov 24
Revision Received:	22nd Nov 24
Editorial Decision:	2nd Dec 24
Revision Received:	4th Dec 24
Accepted:	13th Dec 24

Editor: Esther Schnapp

Transaction Report:

Dear Bo,

Thank you for the transfer of your manuscript to EMBO reports. We received the full set of referee reports that is pasted below and that I shared with you some time ago. Thank you also for sending us a proposed revision plan now.

As I indicated, your manuscript is a borderline case and all referee concerns would need to be successfully addressed in order for us to proceed with your ms here. It seems that you can address the concerns and I would therefore like to give you the opportunity to revise your ms along the lines suggested by the referees. Please co-submit a detailed point-by-point response to all comments with your revised ms.

Acceptance of the manuscript will depend on a positive outcome of a second round of review. It is EMBO reports policy to allow a single round of major revision only and acceptance or rejection of the manuscript will therefore depend on the completeness of your responses included in the next, final version of the manuscript.

We realize that it is difficult to revise to a specific deadline. In the interest of protecting the conceptual advance provided by the work, we recommend a revision within 2 months (10th Nov 2024). Please discuss the revision progress ahead of this time with the editor if you require more time to complete the revisions.

- 1) A data availability section providing access to data deposited in public databases is missing. If you have not deposited any data, please add a sentence to the data availability section that explains that.
- 2) Your manuscript contains statistics and error bars based on $n=2$. Please use scatter blots in these cases. No statistics should be calculated if $n=2$.

3) We replaced Supplementary Information with Expanded View (EV) Figures and Tables that are collapsible/expandable online. A maximum of 5 EV Figures can be typeset. EV Figures should be cited as 'Figure EV1, Figure EV2' etc... in the text and their respective legends should be included in the main text after the legends of regular figures.

5) a complete author checklist, which you can download from our author guidelines <https://www.embopress.org/page/journal/14693178/authorguide>. Please insert information in the checklist that is also reflected in the manuscript. The completed author checklist will also be part of the RPF.

6) Please note that all corresponding authors are required to supply an ORCID ID for their name upon submission of a revised manuscript (<https://orcid.org/>). Please find instructions on how to link your ORCID ID to your account in our manuscript tracking system in our Author guidelines <https://www.embopress.org/page/journal/14693178/authorguide#authorshipguidelines>

7) Before submitting your revision, primary datasets produced in this study need to be deposited in an appropriate public database (see <https://www.embopress.org/page/journal/14693178/authorguide#datadeposition>). Please remember to provide a

reviewer password if the datasets are not yet public. The accession numbers and database should be listed in a formal "Data Availability" section placed after Materials & Method (see also <https://www.embopress.org/page/journal/14693178/authorguide#datadeposition>). Please note that the Data Availability Section is restricted to new primary data that are part of this study. * Note - All links should resolve to a page where the data can be accessed. *

12) All Materials and Methods need to be described in the main text using our 'Structured Methods' format, which is required for all research articles. According to this format, the Methods section includes a separate Reagents and Tools Table (listing key reagents, experimental models, software and relevant equipment and including their sources and relevant identifiers) and a Methods and Protocols section describing the methods using a step-by-step protocol format. The aim is to facilitate adoption of the methodologies across labs. More information on how to adhere to this format as well as a downloadable template (.docx) for the Reagents and Tools Table can be found in our author guidelines: <https://www.embopress.org/page/journal/14693178/authorguide#structuredmethods>.

An example of a Method paper with Structured Methods can be found here: <https://www.embopress.org/doi/full/10.1038/s44320-024-00037-6#sec-4>

You are able to opt out of this by letting the editorial office know (emboreports@embo.org). If you do opt out, the Review Process File link will point to the following statement: "No Review Process File is available with this article, as the authors have

chosen not to make the review process public in this case."

I look forward to seeing a revised form of your manuscript when it is ready.

Kind regards,
Esther

Referee #1:

It is becoming increasingly clear that 3D nuclear reorganization of host genomic DNA is a consequence of viral infection with physiological consequences to disease outcomes. In this report, Wang and colleagues explore how the EBV protein EBNA1 contributes to the immortalization of B cells following EBV infection using a host of multi-omics techniques to do so. The team found that EBNA1 interacts and colocalizes with the chromatin looping factor YY1 to potentially manipulate and reorganize host genomes to facilitate transformation of B cells. The series of experiments performed is quite impressive and include HiChIP, ATAC-seq, ChIP-seq and a number of other multiomic approaches, most of which are combined with functional assays that confirm the -omics data. The work is well done. Nonetheless, the authors hypothesize that since EBNA1 and YY1 sites overlap that EBNA1 might recruit YY1 to EBNA1 sites to induce changes to the 3D looping structure of the host cell DNA. However, the YY1 knock out experiments and data presented in Figure 5 are not convincing that this is the case - and little if no differences are observed EBNA1 loops in the presence or absence of YY1 in these experiments. The authors state that this is likely due to the inefficiency of the YY1 knock-down and do not discuss other potential explanations for these results (which at first glance seem contradictory to the hypothesis). This is a weakness in the manuscript and warrants at a minimum further discussion in the paper with alternative mechanisms for potential alternative mechanisms.

Other minor weaknesses include a lack of detail regarding statistics done using the RNA-seq experiments. None of the relative mRNA graphs have error bars but all have statistical significance indicated

The manuscript has a number of subject-verb agreement mistakes and often "cohesin" is written as "cohesion"

Referee #2:

In this manuscript entitled, "Viral oncogene manipulates YY1 to alter host 3D genome organization and promote B cell proliferation", Wang and colleagues have investigated the role of Epstein-Barr virus (EBV) nuclear antigen leader protein (EBNA1) during the context of infection to better understand how this protein aids to immortalize naïve B lymphocytes (NBLs). The authors have taken a multi-omics approach to address this, and they have assessed promoter-enhancer interactions, gene expression, and transcription factor/histone binding, comparing wild type EBV-infected cells to those in which they have knocked out EBNA1. Overall, the multiomics approaches are sound, though presentation of data could be improved. Additionally, it would be beneficial to confirm some of the key data in follow-up experiments. Finally, the manuscript is a bit hard to follow, and the flow of the narrative and description of key factors could be improved. Specific comments for the authors' consideration are provided.

Major:

1. The expression data for specific genes is simply extrapolated from the larger omics studies. The same can be said for data to support binding of certain factors at specific loci. The authors should consider confirming these results by doing follow-up experiments (infection, RTqPCR for gene expression and ChIP-qPCR for transcription factor binding). This would not only confirm/validate the large-scale analyses, but would also show readers the data repeats (as one would expect error and statistics to be included from at least 3 biological replicates). This seems extremely feasible.
2. The authors state that they generated a CRISPR knockout line for DPF2. However, it is evident from their western blot data (Fig 2E) that the gene is not knocked out. This could be due to a number of reasons, but it seems inappropriate to call these a knockout line based on the immunoblot data. Also, are these the same cells used in Figure 4? It is a bit unclear, but one assumes so. Finally, it is also unclear how this incomplete knockout could impact findings downstream using these cells.
3. A parallel control with the mutant virus would be a nice control to support the data that EBNA1 interacts with YY1 (Fig. 4C).
4. The data showing YY1 depletion and cell viability following CRISPR knockout must be shown. This is a critical transcription factor to cellular homeostasis, and if this protein was successfully knocked out to completion, the data supporting this and cell health absolutely must be included. Without these data, the findings in Fig 5 are difficult to interpret. Similarly, and perhaps this is due to data presentation, but it is unclear what the authors mean in the sentence on lines 283-284 (this should be better articulated/clarified).

5. Figure 6C presents data only for wild-type infection. What does this look like for EBNA1P knockout? This seems important for the overall story/conclusions.
6. The overall conclusion stated on lines 335-338 of the discussion are not entirely supported by the data presented.

Minor:

1. The flow of the manuscript could be improved, especially the introduction and discussion sections (but this also applies to the results section). The text reads very list-like, making it a bit hard to follow. Also, genes/loci/factors are often mentioned but not described (or not described at first mention). Also, for example, sometimes factors/genes/loci are mentioned without rationale, and it would benefit readers to understand these factors to better interpret the data. This makes it difficult for the reader to understand the relevance to the data or the story.
2. The figures are difficult to read. The font is incredibly small in the insets, and the y-axes on many of the omics data are hard to read, making the data difficult for a reader to interpret.
3. Replicate information for all experiments should be clearly articulated (biological and technical replicates), including for western blots and IPs in both the main and supplemental figures.
4. Fig S4C - the data presentation is confusing, and it is unclear what a reader is to gain from this figure as presented. Additionally, the input control needs error bars for the data to be interpretable.
5. Data described in line 225-231 - The authors cite Fig 4A, but it is unclear what a reader should be looking at. Where is the TSS? Perhaps this is due to the figures being too small, and the inset information being impossible to read due to small size.
6. It is not evident what the data in Fig 4G are meant to show. What is on the x-axis? Are these the primers? Also, the data presented in this figure likely should be presented as a bar graph as opposed to a connected line graph (assuming the x-axis represents the primers?).
7. Figure 4 panels seems mislabeled with regards to the text (see lines 272-273). Perhaps a panel was removed?
8. Data in Fig S5 are not referenced in the main manuscript.
9. Data in Fig S6B require densitometry. The effect the authors note is not evident.
10. CTCF looping has also been shown to be important for both HSV and HCMV infection. As this is a herpesvirus manuscript, focusing on EBV, the references supporting these data in other herpesviruses should also be included/referenced (introduction/discussion, where other viruses are mentioned). Similarly, YY1 is an important regulator in HCMV, which also should be included (lines 341-343).

Referee #3:

This manuscript focuses on the mechanism of cell gene regulation by a key viral protein involved in B cell transformation by the cancer-associated Epstein-Barr Virus, the EBV nuclear antigen EBNA-leader protein (EBNA-LP). EBNA-LP has perhaps been the least studied of the nuclear antigens expressed by EBV in latently infected cells, which all have key transcription control functions, as its essential role was initially unclear and it has a complex splicing pattern, with multiple repeated exons giving rise to multiple different length protein forms that differ in their distribution and patterns of expression in both infected cells and cell lines. A number of roles and functions in transcription, often associated with co-regulation with the main EBV transactivator, EBNA-2 have been attributed to EBNA-LP, but some of these roles, based on transfection studies seem to be hard to marry with observations in more physiological settings and using more sophisticated analyses with viral knock-outs and mutants and genome-wide transcriptomics, ChIP etc. This work examines the impact of EBNA-LP on cell gene expression and chromosome looping interactions involving promoters and enhancers and identifies potential new cellular partners through which gene control and chromosome interactions are regulated: the looping factor YY1 and the SWI/SNF component DPF2, although the dual relevance and relationship between and mechanism of interconnectivity (if any) between these two associations is not explored. They are presented quite separately. Overall, there are some interesting observations and new information of relevance to EBV researchers, pointing to potential mechanisms, but there are areas where data could be more robust and the functional details of a role in the mode of action for EBNA-LP for activated and repressed genes seems unclear. There is also no examination of any role in co-regulation with EBNA 2, this is important given that EBNA-LP binding in the B cell genome overlaps with EBNA-2 extensively and there are many common regulated genes. EBNA-2 is absent from the model in Figure 7.

Major points:

1. There is no information on the EBNA-LP KO virus used. The text cites previously published knockouts/mutants but does not state whether these or another has been provided and used in this study? How is it mutated/deleted and has it been checked and characterised?
2. Were the B cell samples analysed for RNA-seq and Hi-ChIP from the same donor(s) and carried out in parallel? This is important to know for comparison of data. There seems to be much more variation in the PCA for the EBNA-LP KO samples-is there a reason for this? Are the genome browser data shown an average or a representative sample? How are the differential genes identified- which sample is compared with which? Uninfected vs wt and LPKO separately or wt vs LPKO. It is not clear from the text, figures or legends and impacts on interpretation.
3. There is novelty in the interacting proteins identified, but further information on interactions and controls are needed. DPF2 (a SWI/SNF component) came from an EBNA-LP pull down and mass spec analysis, but data aren't shown. Why was this protein chosen for follow up? What other factors were pulled down- any other SWI/SNF components? Data or a reference should be provided. Do other SWI/SNF components co-IP in follow up experiments (Fig 2D)? The IgG control here is quite smeary and

- unconvincing. The IP controls should include other nuclear proteins as negative controls to prove specificity- also applies to the YY1 co-IP in Figure 4A. Does the reverse IP for DPF2 pull down EBNA-LP? How does this interaction fit with the known association of EBNA-2 with SWI/SNF factor BRG1. Is EBNA 2 or BRG1 part of the complex and is YY1 in the DPF2 complex and is DPF2 in the YY1 complex?
4. IF images are as expected, simply showing nuclear colocalization of transcription regulators (DPF2, YY1, EBNA-LP). They do not add evidence of co-association.
 5. YY1 knockdown is quoted as 50% but data are not shown. There are minimal effects on reduction of looping from knockdown (13%) which is suggested to be because of poor knock-down. Better knock-down needs to be achieved, whilst taking care to avoid toxicity.
 6. Different examples of genes are used throughout which is quite confusing. Is there a reason for this. Can there be consistent analysis of a set of representative genes for all datasets? Similarly, different cell lines are used for follow-up experiments (e.g. IB4, GM12878). Why are different cell lines used and how do they differ- do they all express the same EBNA-LP forms and are all the relevant candidate genes expressed/repressed.
 7. The conclusion on line 228-233 on impact of EBNA-LP on YY1 binding is not clear from the figure- a zoom in and some quantitation of reads is needed.
 8. How are 'unique' sites identified in each sample? Line 237-241. The conclusion here on the number of HiChIP links in Wt and EBNA-LP at YY1 'unique' sites seems expected from their uniqueness in each sample. What do the authors think can be implied/concluded from these observations?
 9. Line 256- 'DPF2 tether EBNA-LP to chromatin'. I am not convinced this can be concluded. DPF2 may be present and associated with EBNA-LP but this may not be a tethering function.
 10. Involvement of HDAC4 is speculative and should form part of the discussion, not results, unless followed up.

Minor points:

1. The authors state that they used ChatGPT for proofreading. There remain some issues with English throughout (word endings including plurals, meaning of some sentences and typographical errors), so further proofreading is required. There is use of non-scientific terms like 'greatly' or 'massively' to describe magnitudes of changes/effects which should be replaced with numerical values or better descriptors. In places, the introduction reads as a list of facts with no connecting links.
2. The abstract mentions DPF2 but with no information on what this factor is.

We thank the reviewers for their time and efforts to make our manuscript better suited for publication.

Referee 1:

It is becoming increasingly clear that 3D nuclear reorganization of host genomic DNA is a consequence of viral infection with physiological consequences to disease outcomes. In this report, Wang and colleagues explore how the EBV protein EBNA1 contributes to the immortalization of B cells following EBV infection using a host of multi-omics techniques to do so. The team found that EBNA1 interacts and colocalizes with the chromatin looping factor YY1 to potentially manipulate and reorganize host genomes to facilitate transformation of B cells. The series of experiments performed is quite impressive and include HiChIP, ATAC-seq, ChIP-seq and a number of other multiomic approaches, most of which are combined with functional assays that confirm the -omics data. The work is well done. Nonetheless, the authors hypothesize that since EBNA1 and YY1 sites overlap that EBNA1 might recruit YY1 to EBNA1 sites to induce changes to the 3D looping structure of the host cell DNA. However, the YY1 knock out experiments and data presented in Figure 5 are not convincing that this is the case - and little if no differences are observed EBNA1 loops in the presence or absence of YY1 in these experiments. The authors state that this is likely due to the inefficiency of the YY1 knock-down and do not discuss other potential explanations for these results (which at first glance seem contradictory to the hypothesis). This is a weakness in the manuscript and warrants at a minimum further discussion in the paper with alternative mechanisms for potential alternative mechanisms.

We apologize for not making Figure 5 description clearer in the text. The bottom part for each panel represents Hi-ChIP links between enhancer-enhancer and enhancer promoters. Each arc represents a valid genomic interaction. The total number of arcs in each panel represents enhancer-enhancer and enhancer-promoter interaction frequency. The numbers of arcs in EBNA1 knock out virus infected cells and YY1 knock out cells were lower than wt EBV infected cells and YY1 knock out cells in the 4 indicated loci, supporting the notion that EBNA1 recruited YY1 facilitates genomic interactions. We also noted that EBNA1 knockout and YY1 knockout did not completely eliminate loopings at these loci, probably due to the presence of other looping factors such as CTCF. We have added more discussion on alternatives.

Other minor weaknesses include a lack of detail regarding statistics done using the RNA-seq experiments. None of the relative mRNA graphs have error bars but all have statistical significance indicated

The differentially expressed genes from RNA-seq data were identified using DeSeq2. Only significantly altered genes are shown in the figure. The output file from DeSeq doesn't contain error bars.

The manuscript has a number of subject-verb agreement mistakes and often "cohesin" is written as "cohesion"

Thank you for pointing this out. We have corrected these in the manuscript.

Referee 2:

In this manuscript entitled, "Viral oncogene manipulates YY1 to alter host 3D genome organization and promote B cell proliferation", Wang and colleagues have investigated the role of Epstein-Barr virus (EBV) nuclear antigen leader protein (EBVLP) during the context of infection to better understand how this protein aids to immortalize naïve B lymphocytes (NBLs). The authors have taken a multi-omics approach to address this, and they have assessed promoter-enhancer interactions, gene expression, and transcription factor/histone binding, comparing wild type EBV-infected cells to those in which they have knocked out EBVLP. Overall, the multiomics approaches are sound, though presentation of data could be improved. Additionally, it would be beneficial to confirm some of the key data in follow-up experiments. Finally, the manuscript is a bit hard to follow, and the flow of the narrative and description of key factors could be improved. Specific comments for the authors' consideration are provided.

Major:

1. The expression data for specific genes is simply extrapolated from the larger omics studies. The same can be said for data to support binding of certain factors at specific loci. The authors should consider confirming these results by doing follow-up experiments (infection, RTqPCR for gene expression and ChIP-qPCR for transcription factor binding). This would not only confirm/validate the large-scale analyses, but would also show readers the data repeats (as one would expect error and statistics to be included from at least 3 biological replicates). This seems extremely feasible.

We have added qRT-PCR to confirm RNA-seq results (Figure EV1I). They were all done in at least biological triplicates. However, it is a technical challenge to perform ChIP-qPCR, as we do not have access to a sonicator that can handle small numbers of cells (our sonicator can only handle large amount of cells). It is difficult to purify enough naïve B cells and generate enough EBV to infect a large amount of cells.

2. The authors state that they generated a CRISPR knockout line for DPF2. However, it is evident from their western blot data (Fig 2E) that the gene is not knocked out. This could be due to a number of reasons, but it seems inappropriate to call these a knockout line based on the immunoblot data. Also, are these the same cells used in Figure 4? It is a bit unclear, but one assumes so. Finally, it is also unclear how this incomplete knockout could impact findings downstream using these cells.

We changed knockout to depletion, as most people use knock down for siRNA or CRISPRi. Yes, we used the same cell line (IB4 LCLs) for both Figure 4 and Figure 2E. However, as we performed these experiments at different times, we did not use the same batch of DPF2 knockdown cells for experiments. Although DPF2 was not completely depleted, ~70% (according to western blots) depletion of DPF2 was already sufficient to significantly reduce EBNALP and YY1 DNA binding.

3. A parallel control with the mutant virus would be a nice control to support the data that EBNALP interacts with YY1 (Fig. 4C).

Thank you for pointing this out. We agree that using the mutant virus to infect naïve B cells followed by YY1 and EBNALP precipitation would serve as a good negative control. However, the amount of naïve B cells extracted from blood is too few for immunoprecipitation. In our immunoprecipitation experiments, IgG control did not pull down YY1 or EBNALP, which also serves as a good negative control to support the EBNALP and YY1 interaction.

4. The data showing YY1 depletion and cell viability following CRISPR knockout must be shown. This is a critical transcription factor to cellular homeostasis, and if this protein was successfully knocked out to completion, the data supporting this and cell health absolutely must be included. Without these data, the findings in Fig 5 are difficult to interpret. Similarly, and perhaps this is due to data presentation, but it is unclear what the authors mean in the sentence on lines 283-284 (this should be better articulated/clarified).

Thank you for the suggestion. YY1 is essential for LCL cell survival, knock out of YY1 leads to gradually LCL cell death. We have included this data in the figure EV4D and EV4E. We have modified the text on line 283-284 to clarify.

5. Figure 6C presents data only for wild-type infection. What does this look like for EBNALP knockout? This seems important for the overall story/conclusions.

Actually comparing LP repressed sites with the rest of the genome from the same LCL cells are a better comparison than comparing with LP knock out, as these cells have the same growing property.

6. The overall conclusion stated on lines 335-338 of the discussion are not entirely supported by the data presented.

We have modified the text to clarify.

Minor:

1. The flow of the manuscript could be improved, especially the introduction and discussion sections (but this also applies to the results section). The text reads very list-like, making it a bit hard to follow. Also, genes/loci/factors are often mentioned but not described (or not described at first mention). Also, for example, sometimes factors/genes/loci are mentioned without rationale, and it would benefit readers to understand these factors to better interpret the data. This makes it difficult for the reader to understand the relevance to the data or the story.

Thank you for pointing this out. We have made changes in the text.

2. The figures are difficult to read. The font is incredibly small in the insets, and the y-axes on many of the omics data are hard to read, making the data difficult for a reader to interpret.

We have changed the font size and we believe now it is easy to follow.

3. Replicate information for all experiments should be clearly articulated (biological and technical replicates), including for western blots and IPs in both the main and supplemental figures.

We have made clarifications.

4. Fig S4C - the data presentation is confusing, and it is unclear what a reader is to gain from this figure as presented. Additionally, the input control needs error bars for the data to be interpretable.

The data is from a previously published genome-wide CRISPR screen that identified genes essential for the survival of LCL and P3HR1 cells. We selectively extracted the CCND2 sgRNA abundance from this screen data. The sgRNA level corresponds to the extent to which the sgRNA target gene affects cell survival. The low level of CCND2 sgRNA from GM12878 cells suggests that CCND2 is essential for LCL cell survival but not for P3HR1 cells. As there were no replicates from the input, no error bars are shown.

5. Data described in line 225-231 - The authors cite Fig 4A, but it is unclear what a read should be looking at. Where is the TSS? Perhaps this is due to the figures being too small, and the inset information being impossible to read due to small size.

Thank you for pointing this. We labeled CCND2 and CD58 gene body with yellow color. The TSS is located upstream of CCND2 gene body. To make it easy to follow we have added this description in the text.

6. It is not evident what the data in Fig 4G are meant to show. What is on the x-axis? Are these the primers? Also, the data presented in this figure likely should be presented as a bar graph as opposed to a connected line graph (assuming the x-axis represents the primers?).

The x-axis denotes the enhancer region (according to H3K27ac peaks) tested for H3K27ac ChIP-qPCR. Sorry for the confusion, we have changed the P to E (E representing enhancer) for easy understanding.

7. Figure 4 panels seems mislabeled with regards to the text (see lines 272-273). Perhaps a panel was removed?

Thank you for pointing this. We have corrected this.

8. Data in Fig S5 are not referenced in the main manuscript.

Added.

9. Data in Fig S6B require densitometry. The effect the authors note is not evident.

Added.

10. CTCF looping has also been shown to be important for both HSV and HCMV infection. As this is a herpesvirus manuscript, focusing on EBV, the references supporting these data in other herpesviruses should also be included/referenced (introduction/discussion, where other viruses are mentioned). Similarly, YY1 is an important regulator in HCMV, which also should be included (lines 341-343).

Included.

Referee 3:

This manuscript focuses on the mechanism of cell gene regulation by a key viral protein involved in B cell transformation by the cancer-associated Epstein-Barr Virus, the EBV nuclear antigen EBNA-leader protein (EBNA-LP). EBNA-LP has perhaps been the least studied of the nuclear antigens expressed by EBV in latently infected cells, which all have key transcription control functions, as its essential role was initially unclear and it has a complex splicing pattern, with multiple repeated exons giving rise to multiple different length protein forms that differ in their distribution and patterns of expression in both infected cells and cell lines. A number of roles and functions in transcription, often associated with co-regulation with the main EBV transactivator, EBNA-2 have been attributed to EBNA-LP, but some of these roles, based on transfection studies seem to be hard to marry with observations in more physiological settings and using more sophisticated analyses with viral knock-outs and mutants and genome-wide transcriptomics, ChIP etc. This work examines the impact of EBNA-LP on cell gene expression and chromosome looping interactions involving promoters and enhancers and identifies potential new cellular partners through which gene control and chromosome interactions are regulated: the looping factor YY1 and the SWI/SNF component DPF2, although the dual relevance and relationship between and mechanism of interconnectivity (if any) between these two associations is not explored. They are presented quite separately. Overall, there are some interesting observations and new information of relevance to EBV researchers, pointing to potential mechanisms, but there are areas where data could be more robust and the functional details of a role in the mode of action for EBNA-LP for activated and repressed genes seems unclear. There is also no examination of any role in co-regulation with EBNA 2, this is important given that EBNA-LP binding in the B cell genome overlaps with EBNA-2 extensively and there are many common regulated genes. EBNA-2 is absent from the model in Figure 7.

Major points:

1. There is no information on the EBNA-LP KO virus used. The text cites previously published knockouts/mutants but does not state whether these or another has been provided and used in this study? How is it mutated/deleted and has it been checked and characterised?

Thank you for pointing that out. The EBNA-LP KO virus was generously provided by Dr. Wolfgang Hammerschmidt from the German Research Center for Environmental Health and the German Center for Infection Research (DZIF), Munich, Germany. The EBNA-LP mutant virus was created by introducing in-frame translational stop codons in each W2 exon of EBNA-LP (see reference DOI: <https://doi.org/10.1128/mbio.01723-19>). The EBNA-LP mutant bacmids were meticulously scrutinized for any sequence alterations using multiple restriction enzymes and extensive DNA sequencing, confirming the genetic compositions of the pair of maxi-EBV plasmids

2. Were the B cell samples analysed for RNA-seq and Hi-ChIP from the same donor(s) and carried out in parallel? This is important to know for comparison of data. There seems to be much more variation in the PCA for the EBNA-LP KO samples-is there a reason for this? Are the genome browser data shown an average or a representative sample? How are the differential genes identified- which sample is compared with which? Uninfected vs wt and LPKO separately or wt vs LPKO. It is not clear from the text, figures or legends and impacts on interpretation.

RNA-seq and Hi-ChIP data were obtained from the same donors. The variation observed in the EBNALP-KO samples is indeed greater than that in the wild-type samples. We are uncertain about the cause of this increased variation; it may be due to sample handling or increased cellular heterogeneity after LPKO virus infection of naïve B cells compared to the wild-type. The genome browser data shown is from a representative sample. Differential genes were identified by comparing wild-type with LPKO samples. We have added this information to the text.

3. There is novelty in the interacting proteins identified, but further information on interactions and controls are needed. DPF2 (a SWI/SNF component) came from an EBNA-LP pull down and mass spec analysis, but data aren't shown. Why was this protein chosen for follow up? What other factors were pulled down- any other SWI/SNF components? Data or a reference should be provided. Do other SWI/SNF components co-IP in follow up experiments (Fig 2D)? The IgG control here is quite smeary and unconvincing. The IP controls should include other nuclear proteins as negative controls to prove specificity- also applies to the YY1 co-IP in Figure 4A. Does the reverse IP for DPF2 pull down EBNA-LP? How does this interaction fit with the known association of EBNA-2 with SWI/SNF factor BRG1. Is EBNA 2 or BRG1 part of the complex and is YY1 in the DPF2 complex and is DPF2 in the YY1 complex?

EBNALP, like other EBNA2s, does not have a DNA binding domain and is tethered to DNA through interactions with DNA binding proteins. When we first generated the LP ChIP-seq profile, we were impressed by the shape of LP peaks. They are very different from other EBNA2s that all have very distinct and sharp peaks while LP peaks were lower and much wider. The sequence specific DNA binding proteins such as RBPJ that tethers EBNA2 to DNA all had very similar narrow sharp peaks. After we got the LP massspec data, we compared the LP peaks with other TF peaks. DPF2 immediately caught our attention. The shapes of DPF2 peaks are almost identical to LP peaks. More than 82% of LP peaks overlap with DPF2 peaks and their peak patterns are strikingly similar (Fig. 2E). We also found other SWI/SNF complex components, such as SMARCA2 and ARID1A, co-precipitating with EBNALP. We performed immunoprecipitation for EBNALP and DPF2, SMARCA2, and ARID1A simultaneously. Only DPF2 was successfully pulled down. Anti DPF2 antibody did not efficiently immunoprecipitate DPF2. In our EBNALP mass spec data, we also found BRG2; however, we did not have the opportunity to validate whether EBNALP directly interacts with BRG2. YY1 and EBNALP association is reported recently by Dr. Luftig's group. EBNA2 and LP ChIP-seq peaks are very different in genomic position (LP 33% vs EBNA2 16% at promoters) and shape. We only focused on sites unique for LP and did not study other sites. We have not tested if YY1 and DPF2 interact directly but probably in the same complex.

4. IF images are as expected, simply showing nuclear colocalization of transcription regulators (DPF2, YY1, EBNA-LP). They do not add evidence of co-association.

IF images are an additional line of evidence supporting the association.

5. YY1 knockdown is quoted as 50% but data are not shown. There are minimal effects on reduction of looping from knockdown (13%) which is suggested to be because of poor knock-down. Better knock-down needs to be achieved, whilst taking care to avoid toxicity.

We have now included the Western blot and cell growth for YY1 depletion (figure EV2E and EV2F). It is very difficult to balance cell viability and depletion efficiency. But at least the trends are consistent with our hypothesis.

6. Different examples of genes are used throughout which is quite confusing. Is there a reason for this. Can there be consistent analysis of a set of representative genes for all datasets? Similarly, different cell lines are used for follow-up experiments (e.g. IB4, GM12878). Why are different cell lines used and how do they differ- do they all express the same EBNA-LP forms and are all the relevant candidate genes expressed/repressed.

The examples were used mostly by their differential expression in the RNA-seq experiments. We also intersected the data with the genome wide CRISPR screen that identified genes essential for LCL growth and survival. We only used LCL lines that have been in culture for long term where LP levels are stable and comparable. Some lines are easier to be transduced by lentiviruses, so we used different lines for this reason.

7. The conclusion on line 228-233 on impact of EBNA-LP on YY1 binding is not clear from the figure- a zoom in and some quantitation of reads is needed.

We have provided zoom in image in Figure EV2D.

8. How are 'unique' sites identified in each sample? Line 237-241. The conclusion here on the number of HiChIP links in Wt and EBNA-LP at YY1 'unique' sites seems expected from their uniqueness in each sample. What do the authors think can be implied/concluded from these observations?

The 'unique' sites refer to YY1 Cut&Run peaks that are significantly higher in either the WT or EBNA-LP KO group. More HiChIP reads detected at these 'unique' YY1 sites from each group suggest that YY1 is involved in these DNA-DNA interactions. The absence of YY1 reduces these DNA-DNA interactions.

9. Line 256- 'DPF2 tether EBNA-LP to chromatin'. I am not convinced this can be concluded. DPF2 may be present and associated with EBNA-LP but this may not be a tethering function.

The striking similarity of DPF2 and LP ChIP-seq peak pattern and overwhelming overlapping between LP peaks with DPF2, DPF2 CRISPR depletion significantly reduces LP DNA binding at all sites evaluated, all strongly support the notion that DPF2 tethers LP to DNA. In our LP masspec, we did not find other DNA binding proteins except the chromatin modifiers that we could not validate the association and DPF2 is known to bind to H3K14ac and H4K16ac.

10. Involvement of HDAC4 is speculative and should form part of the discussion, not results, unless followed up.

HDAC4 part was moved to discussion.

Minor points:

1. The authors state that they used ChatGPT for proofreading. There remain some issues with English throughout (word endings including plurals, meaning of some sentences and typographical errors), so further proofreading is required. There is use of non-scientific terms like 'greatly' or 'massively' to describe magnitudes of changes/effects which should be replaced with numerical values or better descriptors. In places, the introduction reads as a list of facts with no connecting links.

Thanks for the suggestions. We have modified the text and replaced “greatly” and “massively”.

2. The abstract mentions DPF2 but with no information on what this factor is.

Thanks for the suggestions. We have added the information of DPF2 in the abstract.

Dear Dr. Zhao,

We have now received the comments from referees 2 and 3 on your revised ms, I paste both reports below.

As you will see, neither of the referees strongly supports the publication of your revised study, and referee 2 even indicates in the ms summary table that the ms should not be published after revision. It appears that the revised ms was not thoroughly revised and prepared.

However, given the related published paper (doi: 10.1371/journal.ppat.1011950) I assume it is likely that your data are valid. I would therefore like to give you the exceptional opportunity to carefully address the current referee comments and to carefully revise your ms a second time. It would be good to also provide the ChIPqPCR data referee 2 is asking for. Please also discuss the published paper in relation to your data. You do not need to worry about novelty, as your ms was submitted before the other paper was published in a journal. However, it would be good if you could resubmit your ms at the end of November, so that it can still be published online this year, if possible, and if the referees will support its publication.

Please also include a separate Research and Tools table file and a completed author checklist with your next submission. This information was already present in my last decision letter. Please also co-submit a detailed point-by-point response to the referee comments with your next ms version.

Please also address the comments on your figure legends as follows:

- Please note that the figure panels 1c, f is not labelled in the figure. This needs to be rectified.
- Please note that the figure 1d-f is mislabeled as figure 1c-d in the manuscript. This needs to be rectified.
- Please define the annotated p values *** as well as provide the exact p-values for the same in the legend of figure 4a; EV 1e, i; EV 2f; as appropriate.
- Please note that the exact p values are not provided in the legends of figures 1e-f; 2b, e; 3b; 5e, g-h; EV 4a.
- Please indicate the statistical test used for data analysis in the legends of figures 1a, e-f; 3b; 4a; EV4a, d.
- Please note that in figures 5g-h; there is a mismatch between the annotated p values in the figure legend and the annotated p values in the figure file that should be corrected.
- Please note that for the figure EV 4d, p-values and statistical tests are indicated in the legend. However, comparison for the same, ""****"" has not been represented in the figure. Please rectify this in the figure or legend as applicable.
- Please note that information related to n is missing in the legends of figures 1e-f; 2b; 3c; 4b; EV 2c; EV 4a.
- Please note that the error bars are not defined in the legends of figures 1e-f; EV 4a.

Please let me know immediately if you have any questions or comments.

I look forward to seeing a newly revised form of your manuscript as soon as possible.

Referee #2:

The authors have revised their manuscript, yet many of the same initial concerns remain. The prior critiques were largely not addressed, as indicated by the very minimal changes to the text. Some data were added, but the authors resoundingly did not address many points, thus concerns remain. What is also concerning is that the authors indicate they have made the suggested changes, yet more often than not, this is just not the case.

The prior critiques suggested the inclusion of ChIP-PCR data, and it is hard to believe the authors cannot find a sonicator at their institution to perform the experiment. Similarly, ChIPqPCR does not require mass quantities of starting material; this experiment is more than feasible. This response collectively falls short, as ChIP-qPCR data are included in this manuscript elsewhere (see Fig 4).

Data also remain analyzed in singlet (see Fig 1/ Fig 3B/ Fig EV4A, where no error bars exist, yet statistical significance is shown); this is impossible to interpret.

Other points were also not adequately addressed. For example, the manuscript, especially the introduction/discussion are very choppy. The text reads very "list-like" and it is difficult to read/follow.

Similarly, the authors failed to include HSV and HCMV in their discussion on CTCF, which is indicated as addressed in their responses.

Conclusions from original lines 335-338 were not addressed by the authors.

As indicated in prior critiques, IF does not add evidence of co-association. The authors are also incorrect in stating such images provide additional lines of evidence for association. One can conclude proteins co-localize, but not that they associate.

Referee #3:

The authors have addressed my queries to some extent but the responses in the rebuttal do not all indicate that the manuscript text has been amended accordingly as I would have expected. It is very likely that readers will have similar queries. Addressing queries only via a rebuttal will not improve the manuscript. I see that the authors note that a publication from another group since submission (J. M. Cable, N. M. Reinoso-Vizcaino, R. E. White, M. A. Luftig, Epstein-Barr virus protein EBNA-LP engages YY1 through leucine-rich motifs to promote naive B cell transformation. PLoS Pathog 20, 934 e1011950 (2024)) has described an interaction between LP and YY1. This work is cited in the revised manuscript, but how does this affect the novelty of the findings described here?

Major points:

Original query 1: There is no information on the EBNA-LP KO virus used.

Thank you for providing the information. This information needs to be included in the methods or elsewhere. I cannot see that it has been added?

Original query 3: There is novelty in the interacting proteins identified, but further information on interactions and controls are needed.

The response here is helpful and acceptable, but it is not clear that the additional information provided has been incorporated into the manuscript as this is not stated. Please can this be confirmed or added. It is important to address my points including on EBNA2.

Original query 4: IF images are as expected, simply showing nuclear colocalization of transcription regulators (DPF2, YY1, EBNA-LP). They do not add evidence of co-association.

The response here is not satisfactory. Please amend the manuscript to indicate that IF provides evidence of similar nuclear localisation patterns, not evidence of an interaction.

Original query 6: Different examples of genes are used throughout which is quite confusing. Is there a reason for this. Can there be consistent analysis of a set of representative genes for all datasets? Similarly, different cell lines are used for follow-up experiments (e.g. IB4, GM12878).

The explanation for the different genes is fine, but please add information to the text as described in the rebuttal as to why different B cell lines were used (transduction efficiency). This will help the reader.

Original query 9: Line 256- 'DPF2 tether EBNA-LP to chromatin'. I am not convinced this can be concluded. DPF2 may be present and associated with EBNA-LP but this may not be a tethering function.

I do not agree with the response here and the line remains in the manuscript, although in other places it states 'may tether'. I maintain that this line should be modified to be less strong when first stated at least, along the lines of what is described in the rebuttal e.g. 'evidence supports the hypothesis that DPF2 tethers LP to DNA or stabilises its binding'. The absence of finding any other DNA binding proteins, does not add to the strength of argument in my opinion.

We thank the reviewers for their time and efforts to make our manuscript better suited for publication.

Referee #2:

The authors have revised their manuscript, yet many of the same initial concerns remain. The prior critiques were largely not addressed, as indicated by the very minimal changes to the text. Some data were added, but the authors resoundingly did not address many points, thus concerns remain. What is also concerning is that the authors indicate they have made the suggested changes, yet more often than not, this is just not the case.

The prior critiques suggested the inclusion of ChIP-PCR data, and it is hard to believe the authors cannot find a sonicator at their institution to perform the experiment. Similarly, ChIPqPCR does not require mass quantities of starting material; this experiment is more than feasible. This response collectively falls short, as ChIP-qPCR data are included in this manuscript elsewhere (see Fig 4).

We are restricted by the number of naïve B cells we can get from each donor. The number of cells we get is too low to generate reliable ChIP-PCR data. The ChIP-PCR data in the manuscript were generated from cell lines.

Data also remain analyzed in singlet (see Fig 1/Fig 3B/Fig EV4A, where no error bars exist, yet statistical significance is shown); this is impossible to interpret.

The data came from RNA-seq, which was done in triplicates. All the examples are statistically significant. The error bars are now included.

Other points were also not adequately addressed. For example, the manuscript, especially the introduction/discussion are very choppy. The text reads very "list-like" and it is difficult to read/follow.

We have now used Grammarly to correct the writing throughout the whole manuscript, and highlighted the part that needs to be modified with red.

Similarly, the authors failed to include HSV and HCMV in their discussion on CTCF, which is indicated as addressed in their responses.

HSV and CMV are now included.

Conclusions from original lines 335-338 were not addressed by the authors.

CCND2 is essential for cell cycle progression, and BCL2L11, CASP1/5 are genes crucial for cell apoptosis and cell death. Using wt and lpko EBV virus to infect naïve B cells, we demonstrated that EBNALP altered these gene transcription (from RNA-seq), their enhancer-promoter interactions (from HiChIP), and YY1 binding to these gene loci (from YY1 CUT&RUN). In addition, using CRISPRi to perturb the CCND2 enhancers bound with YY1 and where its interaction with CCND2 promoter were affected by EBNALP showed that silencing CCND2 enhancer downregulates CCND2 gene expression. Based on these evidence, we believe our conclusion is reasonable. However, to make it clearer we have changed the text to "alter the expression of genes essential for cell cycle progression and cell death.

”

As indicated in prior critiques, IF does not add evidence of co-association. The authors are also incorrect in stating such images provide additional lines of evidence for association. One can conclude proteins co-localize, but not that they associate.

We changed to colocalization in the description now. The association came from immune precipitation.

Referee #3:

The authors have addressed my queries to some extent but the responses in the rebuttal do not all indicate that the manuscript text has been amended accordingly as I would have expected. It is very likely that readers will have similar queries. Addressing queries only via a rebuttal will not improve the manuscript. I see that the authors note that a publication from another group since submission (J. M. Cable, N. M. Reinoso-Vizcaino, R. E. White, M. A. Luftig, Epstein-Barr virus protein EBNA-LP engages YY1 through leucine-rich motifs to promote naive B cell transformation. PLoS Pathog 20, 934 e1011950 (2024)) has described an interaction between LP and YY1. This work is cited in the revised manuscript, but how does this affect the novelty of the findings described here?

Our manuscript is the first one to describe novel mechanisms through which EBNA-LP regulates gene expression by altering host 3D genome organization and manipulating YY1 DNA binding.

Major points:

Original query 1: There is no information on the EBNA-LP KO virus used.

Thank you for providing the information. This information needs to be included in the methods or elsewhere. I cannot see that it has been added?

It is now included.

Original query 3: There is novelty in the interacting proteins identified, but further information on interactions and controls are needed.

The response here is helpful and acceptable, but it is not clear that the additional information provided has been incorporated into the manuscript as this is not stated. Please can this be confirmed or added. It is important to address my points including on EBNA2.

Additional info added. EBNA2 was not identified in the experiments.

Original query 4: IF images are as expected, simply showing nuclear colocalization of transcription regulators (DPF2, YY1, EBNA-LP). They do not add evidence of co-association.

The response here is not satisfactory. Please amend the manuscript to indicate that IF provides evidence of similar nuclear localisation patterns, not evidence of an interaction.

Corrected as suggested.

Original query 6: Different examples of genes are used throughout which is quite confusing. Is there a reason for this. Can there be consistent analysis of a set of representative genes for all datasets? Similarly, different cell lines are used for follow-up experiments (e.g. IB4, GM12878).

The explanation for the different genes is fine, but please add information to the text as described in the rebuttal as to why different B cell lines were used (transduction efficiency). This will help the reader.

IB4 cells were established many years ago. The EBNA1P expression is very stable in size and EBNA1P ChIP-seq was done in that line. GM12878 LCLs are newer LCLs where EBNA1P size still varies (due to different numbers of W repeats) and as ENCODE tier 1 cell line, there are a lot of sequencing based data available. We have a lot of experience doing CRISPR with GM12878. They have different efficiencies in terms of transduction or transfection. We have added these description to the text.

Original query 9: Line 256- 'DPF2 tether EBNA1P to chromatin'. I am not convinced this can be concluded. DPF2 may be present and associated with EBNA-LP but this may not be a tethering function.

I do not agree with the response here and the line remains in the manuscript, although in other places it states 'may tether'. I maintain that this line should be modified to be less strong when first stated at least, along the lines of what is described in the rebuttal e.g. 'evidence supports the hypothesis that DPF2 tethers LP to DNA or stabilises its binding'. The absence of finding any other DNA binding proteins, does not add to the strength of argument in my opinion.

"May tether" is a substantial tone down version from the original version.

Dear Dr. Zhao,

Thank you for the submission of your newly revised manuscript. We can in principle accept it now, however, a few editorial requests still need to be addressed:

- Please provide up to 5 Keywords with your ms file.
- The GSE210454 dataset is no longer mentioned in the ms, but instead GSE277748 is. Can you please explain why? We need a direct URL for the GSE277748 dataset in the Data Availability section.
- "Declaration of Interests" needs to be renamed to "Disclosure Statement and Competing Interests"
- All Author Contributions need to be removed from the ms. All contributions need to be entered during online ms submission.
- The Reference format needs to be alphabetical, not numerical; et al needs to be used after 10 author names. The EMBO reports reference style can be found in EndNote.
- The AI declaration is provided after the DAS; however, it should be part of the Methods section
- The Figure legends should be placed after the References, at the very end of the ms file.
- We still need a completed author checklist from you. It can be downloaded from our author guidelines <<https://www.embopress.org/page/journal/14693178/authorguide>>. The completed author checklist will also be part of the transparent peer-review file.
- All Funding info and all NIH grant numbers listed in the ms must also be entered during online ms submission.
- All Source data (SD) are missing. Hannah contacted you on the 11th of September asking for source data. Please submit all SD as one SD file per main figure and one SD file for all EV figures with your final ms.

I would like to suggest some changes to the title and abstract that needs to be written in present tense. Please let me know whether you agree with the following:

Viral oncogene EBNALP regulates YY1 DNA binding and alters host 3D genome organization

The Epstein-Barr virus (EBV) nuclear antigen leader protein (EBNALP) is essential for the immortalization of naïve B lymphocytes (NBLs). However, the mechanisms remain elusive. To understand EBNALP's role in B cell transformation, we compare NBLs infected with wild-type EBV and an EBNALP-null mutant EBV using multi-omics techniques. EBNALP inactivation alters enhancer-promoter interactions, resulting in decreased CCND2 and increased CASP1 and BCL2L11 expression. Mechanistically, EBNALP interacts with and colocalizes with the looping factor YY1. Depletion of EBNALP reduces YY1 DNA binding and enhancer-promoter interactions, similar to effects observed with YY1 depletion. Furthermore, EBNALP colocalizes with DPF2, a protein that binds to H3K14ac and H4K16ac. CRISPR depletion of DPF2 reduces both EBNALP and YY1 DNA binding, suggesting that the DPF2/EBNALP complex may tether YY1 to DNA to increase enhancer-promoter interactions. EBNALP inactivation also increases enhancer-promoter interactions at the CASP1 and BCL2L11 loci, along with elevated DPF2 and YY1 binding and DNA accessibility. Our data suggest that EBNALP regulates YY1 to rewire the host genome, which might facilitate naïve B cell transformation.

EMBO press papers are accompanied online by A) a short (1-2 sentences) summary of the findings and their significance, B) 2-3 bullet points highlighting key results and C) a synopsis image that is exactly 550 pixels wide and 200-600 pixels high (the height is variable). The synopsis image should provide a sketch of the major findings, like a graphical abstract. Please note that text needs to be readable at the final size. Please send us this information along with the final manuscript.

The authors have addressed all minor editorial requests.

Bo Zhao
Harvard Medical School
United States

Dear Dr. Zhao,

I am very pleased to accept your manuscript for publication in the next available issue of EMBO reports. Thank you for your contribution to our journal.

Yours sincerely,
